# ADAPTIVE REGULARIZATION OF REPRESENTATION RANK AS AN IMPLICIT CONSTRAINT OF BELLMAN EQUATION

**Qiang He**[1]**, Tianyi Zhou**[2]**, Meng Fang**[3]**, Setareh Maghsudi**[1]
[1]Ruhr University Bochum, [2]University of Maryland, College Park, [3]University of Liverpool
{Qiang.He, Setareh.Maghsudi}@rub.de, tianyi@umd.edu, Meng.Fang@liverpool.ac.uk

## ABSTRACT

Representation rank is an important concept for understanding the role of Neural Networks (NNs) in Deep Reinforcement learning (DRL), which measures the expressive capacity of value networks. Existing studies focus on unboundedly maximizing this rank; nevertheless, that approach would introduce overly complex models in the learning, thus undermining performance. Hence, fine-tuning representation rank presents a challenging and crucial optimization problem. To address this issue, we find a guiding principle for adaptive control of the representation rank. We employ the Bellman equation as a theoretical foundation and derive an upper bound on the cosine similarity of consecutive state-action pairs representations of value networks. We then leverage this upper bound to propose a novel regularizer, namely BEllman Equation-based automatic rank Regularizer (BEER). This regularizer adaptively regularizes the representation rank, thus improving the DRL agent's performance. We first validate the effectiveness of automatic control of rank on illustrative experiments. Then, we scale up BEER to complex continuous control tasks by combining it with the deterministic policy gradient method. Among 12 challenging DeepMind control tasks, BEER outperforms the baselines by a large margin. Besides, BEER demonstrates significant advantages in Q-value approximation. Our code is available at https://github.com/sweetice/BEER-ICLR2024.

## 1 INTRODUCTION

Deep reinforcement learning (DRL), empowered by the large capacity of neural networks (NNs), has made notable strides in a variety of sequential decision-making tasks, ranging from games (Mnih et al., 2015; Silver et al., 2018; Vinyals et al., 2019), to robotics control (Haarnoja et al., 2018), and large language models (Christiano et al., 2017; OpenAI, 2023). Yet, despite these advancements, NNs within DRL systems are still largely treated as black-box function approximators (Schulman et al., 2017; Lillicrap et al., 2016; Fujimoto et al., 2018; Haarnoja et al., 2018). Investigating the specific roles played by NNs in DRL may offer insights into their function, enabling further optimization of DRL agents from the aspect of NNs. Recent work (Kumar et al., 2021; Lyle et al., 2022; He et al., 2023a) has begun to shed light on the role of NNs in RL, proposing to leverage some properties (e.g., capacity (Lyle et al., 2022) and plasticity (Lyle et al., 2023; Nikishin et al., 2022; Sokar et al., 2023)) of NNs to improve DRL agents.

A core concept to study the properties of NNs in the DRL setting is the representation rank (Kumar et al., 2021). It measures the representation capacity of a NN by performing a singular value decomposition on the output of its representation layer, which serves as the input representation for value functions. Previous studies (e.g., InFeR (Lyle et al., 2022) and DR3 (Kumar et al., 2021)) heavily focus on unbounded maximizing this metric. Yet, that approach might cause overfitting by constructing an overly complex function approximation, thus inhibiting the model's ability to generalize to new data. The resulting complexity demands more data and makes the model susceptible to noise that deters sampling efficiency and robustness in RL. As such, an indiscriminate maximization of representation rank is not advisable. A more desirable method is to *adaptively* control the representation rank. However, empirically fine-tuning the balance of representation rank can be tricky. An excessively complex model induces the issues mentioned above, whereas overly

simple models lose the capability to obtain the optimal policy. As a result, it is imperative to search for a guiding principle that allows for the adaptive control of the representation rank.

A primary discovery of this paper is that an adaptive control of the representation rank can be derived from Bellman equation (Sutton & Barto, 2018) as an implicitly imposed constraint on NNs. Specifically, this can be achieved by an upper bound on the cosine similarity between the representations of consecutive state-action pairs, determined by factors like the discount factor, the weights of the last layer of the neural network, and the representations norms. Cosine similarity measures the linear dependent relationship between the composition vectors, thus affecting the rank. The implication is profound: The cosine similarity is constrained, thereby restraining the representation rank itself (Kumar et al., 2021; Lyle et al., 2022; 2023). However, a previous work (Kumar et al., 2021) shows that the dynamics of NNs result in feature co-adaptation, which potentially makes DRL agents fail to hold the bound. Thus, these derived upper bounds provide a criterion to adaptively control the representation rank. Motivated by these findings, we introduce a novel regularizer, namely BEllman Equation-based automatic rank Regularizer (BEER), designed to control the representation rank by regularizing the similarity between adjacent representations of the value network. That allows DRL agents to **adaptively** control the rank based on the constraint of the Bellman equation and preserve desirable representation ranks. Specifically, the BEER regularizer only reduces the similarity (thereby increasing the representation rank) when the similarity exceeds the upper bound. Numerical experiments, e.g., in Figure 1, show that our method outperforms existing approaches that do not control or overly strengthen the representation rank. For instance, we demonstrate that algorithms like InFeR, which keep maximizing the representation rank without constraint, produce high approximation errors. In contrast, our proposed regularizer, BEER, functions significantly better by adaptively regularizing the representation rank. We further scale up BEER to challenging continuous control suite DMcontrol (Tunyasuvunakool et al., 2020), where BEER performs better than the existing methods DR3 and InFeR.

Our main contribution is three folds:

1. We discover the implicit constraints on representations of the Bellman equation and establish an upper bound on the cosine similarity.

2. We find a theoretical principle to maintain the representation rank adaptively. We design a novel, theoretically-grounded regularizer called BEER. It controls the representation rank by adhering to the constraints imposed by the Bellman equation.

3. The empirical experiments validate that the BEER regularizer can help models maintain a balanced representation rank thereby further enhancing the DRL agents' performance. We scale up BEER to 12 challenging continuous control tasks, and BEER outperforms existing approaches by a large margin.

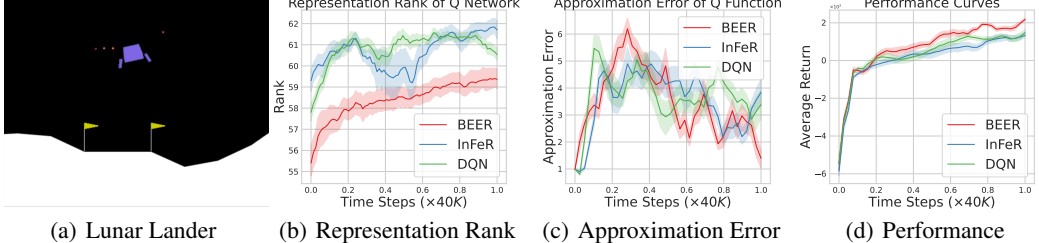

(a) Lunar Lander    (b) Representation Rank    (c) Approximation Error    (d) Performance

Figure 1: Illustrative experiments on the Lunar Lander environment, with results averaged over ten random seeds. The shaded area represents half a standard deviation. (a) A snapshot of the Lunar Lander environment. (b) Comparison of representation ranks. BEER exhibits a more balanced rank compared to InFeR and DQN. (c) Approximation errors of different algorithms. BEER displays a lower approximation error compared to both DQN and InFeR in the latter stage (0.9 to $1 \times 40K$ time steps). (d) Performance curves substantiating the superiority of BEER.

## 2   PRELIMINARIES

This paper employs a six-tuple Markov Decision Process (MDP) $(\mathcal{S}, \mathcal{A}, R, P, \gamma, \rho_0)$ to formalize RL, where $\mathcal{S}$ represents a state space, $\mathcal{A}$ denotes an action space, $R : \mathcal{S} \times \mathcal{A} \to \mathbb{R}$ a reward function,

$P : \mathcal{S} \times \mathcal{A} \to p(s)$ is a transition kernel, $\gamma \in [0, 1)$ serves as a discount factor, and $\rho_0$ specifies an initial state distribution. The objective of RL is to optimize the policy through return, defined as $R_t = \sum_{i=t}^{T} \gamma^{i-t} r(s_i, a_i)$. The action value ($Q$) function measures the quality of an action $a$ given a state $s$ for a policy $\pi$, defined as

$$Q^\pi(s, a) = \mathbb{E}_{\tau \sim \pi, p}[R_\tau | s_0 = s, a_0 = a], \tag{1}$$

where $\tau$ is a state-action sequence $(s_0, a_0, s_1, a_1, s_2, a_2 \cdots)$ generated by the policy $\pi$ and transition probability $P$. The $Q$ value adheres to the Bellman equation (Sutton & Barto, 2018):

$$Q^\pi(s, a) = r(s, a) + \gamma \mathbb{E}_{s', a'}[Q^\pi(s', a')], \tag{2}$$

where $s' \sim P(\cdot | s, a)$ and $a \sim \pi(\cdot | s)$. In scenarios where the state or action space is prohibitively large, leveraging conventional tabular methods for value storage becomes computationally intractable. Function approximations such as NNs are commonly employed for value approximation in these cases.

It is instructive to adopt a matrix view for elaborating on the MDP (Agarwal et al., 2019). Let $\mathbf{Q}^\pi \in \mathbb{R}^{(|\mathcal{S}| \cdot |\mathcal{A}|) \times 1}$ denote all Q values, and $\mathbf{r}$ as vectors of the same shape. We extend notation $P^\pi$ to represent a matrix of dimension $(|\mathcal{S}| \cdot |\mathcal{A}|) \times (|\mathcal{S}| \cdot |\mathcal{A}|)$, where each entry is $P^\pi_{(s,a),(s',a')} :=$ $P(s'|s, a)\pi(a'|s')$, where $P^\pi$ is induced by a stationary policy $\pi$. It is straightforward to verify the matrix form of the Bellman equation (Agarwal et al., 2019)

$$\mathbf{Q}^\pi = \mathbf{r} + \gamma P^\pi \mathbf{Q}^\pi. \tag{3}$$

**Representations of value functions.** In the DRL setting, we express the $Q$ function as $Q^\pi(s, a) = \phi^\top(s, a)w$, where $\phi(s, a) \in \mathbb{R}^{N \times 1}$ denotes a feature representation of dimension N and $w \in \mathbb{R}^{N \times 1}$ is a weight vector. Note that $w$ is not related to the input pair $(s, a)$. In this setting, $\phi(s, a)$ can be considered as the output of the penultimate layer of a neural network, and $w$ corresponds to the weight of the network's final layer. To assess the expressiveness of the value function representation, we adopt the concept of the "representation rank" as introduced by Lyle et al. (2022). The representation rank is formally defined as follows.

**Definition 1** (Numerical representation rank). Let $\phi : \mathcal{S} \times \mathcal{A} \mapsto \mathbb{R}^N$ be a representation mapping. Let $X_n \subset \mathcal{S} \times \mathcal{A}$ be a set of $n$ state-action pairs samples drawn from a fixed distribution $p$. The representation rank of the value function is defined as

$$\text{Rank}(\phi, X_n; \epsilon, n) = \mathbb{E}_{X_n \sim p}\big|\{\sigma \in \text{Spec}\phi(\frac{1}{\sqrt{n}}X_n) : |\sigma| > \epsilon\}\big|, \tag{4}$$

where Spec denotes the singular spectrum containing the singular values, produced by performing a singular value decomposition on matrix $\phi(X_n)$.

We set $p$ to be a uniform distribution over $\mathcal{S} \times \mathcal{A}$ for the scope of this paper. In the context of a finite state space $X_n$ and $\epsilon = 0$, the numerical representation rank —hereafter referred to as the "representation rank"— corresponds to the dimension of the subspace spanned by the representations. Selecting $\epsilon > 0$ filters out the small singular values. Therefore, one can determine the representation rank by calculating the singular spectrum of a representation matrix, which is constructed by concatenating the sampled state-action pairs

## 3 METHOD

In pursuit of a principled approach to regulating the representation rank, we deviate from the common practice of directly manipulating phenomena such as primacy bias (Nikishin et al., 2022) or dormant neurons (Sokar et al., 2023) within NNs. While such heuristics offer valuable insights, they seldom provide generalizable theoretical underpinnings. One of the challenges in deriving theoretical insights from NNs is their inherent flexibility, which often leaves little room for rigorous analysis. To circumvent this limitation, we incorporate the Bellman equation in the context of NNs for DRL. The intersection of Bellman equations and NNs offers a fertile analysis ground, thereby facilitating the extraction of solid guarantees. We commence our discussion with an analytical examination of the Bellman equation and then derive an upper bound for the inner product of two consecutive representations of value networks. This upper bound is immediately transformable into a new

bound that constrains the cosine similarity of the representations. Notably, the cosine similarity implicitly restricts the representation rank. Motivated by the theoretical analysis, we introduce the BEER regularizers to adaptively control the representation rank. We consider a value function approximation problem with constraints. Then the BEER regularizer serves as a penalty to solve the optimization problem. Contrary to methods that unboundedly maximize the representation rank (Lyle et al., 2022; Kumar et al., 2021; He et al., 2023a), our strategy imposes a well-calibrated constraint, which is instrumental in achieving a more nuanced and effective regularization.

## 3.1 UPPER BOUNDS ON SIMILARITY MEASURES FOR ADJACENT REPRESENTATIONS

We examine the Bellman equation through the interplay between Q values and their underlying representations within DRL. The view based on the Q value can be described as an inner product between the representation vector and a corresponding weight vector. This allows us to rewrite the Bellman equation as

$$\Phi^\top w = \mathbf{r} + \gamma P^\pi \Phi^\top w. \tag{5}$$

Inspired by Equation (3), we isolate the term involving the representation $\Phi$ on one side of the equation. This manipulation yields an inter-temporal relationship between $\Phi$ and its successor, i.e.,

$$(\Phi^\top - \gamma P^\pi \Phi^\top)w = \mathbf{r}. \tag{6}$$

Taking $L_2$-norm of both sides results in

$$\|(\Phi^\top - \gamma P^\pi \Phi^\top)w\|_2 = \|\mathbf{r}\|_2. \tag{7}$$

Given the intractability of finding an optimal policy in DRL if either the representation function or the weight vector becomes trivial (e.g., zero vector), we introduce the following assumption.

**Assumption 1.** Both the representation function $\phi$ and the weight vector $w$ are non-trivial; specifically, $\phi$ is not a constant function, and $w$ is not a zero vector.

With Assumption 1 and by utilizing the definition of the operator norm (Debnath & Mikusinski, 2005), we arrive at

$$\|\Phi^\top - \gamma P^\pi \Phi^\top\|_{op} \geq \frac{\|\mathbf{r}\|_2}{\|w\|_2}, \tag{8}$$

where $\|\cdot\|_{op}$ symbolizes the operator norm. Equation (8) shows an implicit requirement of the representation $\Phi$ originating from the Bellman equation, which inspires us to re-consider the Bellman equation in Equation (2). We derive the following theorem constraining the inner product of two adjacent representations of value functions.

**Theorem 1.** *Under Assumption 1, given Q value $Q(s,a) = \phi(s,a)^\top w$,*

$$\langle \phi(s,a), \overline{\phi(s',a')} \rangle \leq (\|\phi(s,a)\|^2 + \gamma^2 \|\overline{\phi(s',a')}\|^2 - \frac{\|r\|^2}{\|w\|^2}) \frac{1}{2\gamma}, \tag{9}$$

*where $\langle \cdot, \cdot \rangle$ represents the inner product, $\phi$ is a state-action representation vector, $w$ is a weight, and $\overline{\phi(s',a')} = \mathbb{E}_{s',a'}\phi(s',a')$ denotes the expectation of the representation of the next state action pair.*

To keep the consistency, we defer all proofs to the Appendix. Theorem 1 states the relationship of representations at adjacent time steps. We derive the theorem from the Bellman equation as a necessary condition for optimality associated with the optimal value function. Hence, the upper bound in Equation (9) is also an interesting result concerning the optimal value networks. We can directly obtain the cosine similarity between the current representation and the one at the next step by dividing both sides of the Equation (9) by their norms.

*Remark* 1. Under Assumption 1, given Q value $Q(s,a) = \phi(s,a)^\top w$, then

$$\cos(\phi(s,a), \overline{\phi(s',a')}) \leq (\|\phi(s,a)\|^2 + \gamma^2 \|\overline{\phi(s',a')}\|^2 - \frac{\|r\|^2}{\|w\|^2}) \frac{1}{2\gamma \|\phi(s,a)\| \|\overline{\phi(s',a')}\|}. \tag{10}$$

Remark 1 discusses the cosine similarity between two adjacent representations of the value function. The representation rank, as defined in Definition 1, is determined by computing the singular spectrum, which quantifies the number of filtered singular values. Under these conditions, the cosine similarity

is intrinsically linked to the representation rank. For example, if all vectors in the representation matrix are orthogonal cosine similarity is zero, then they are linearly independent. That means that the matrix is full rank. If some vectors of the representation matrix are linearly dependent, which means the cosine similarity taking 1, then they will contribute to a lower rank. Explicitly controlling cosine similarity allows for adaptive adjustment of the representation rank, a methodology also echoed in prior research (Kumar et al., 2021; Lyle et al., 2022).

### 3.2 ADAPTIVE REGULARIZATION OF REPRESENTATION RANK

Previous work (e.g., InFeR, DR3) generally unboundedly maximizes the representation rank by regularizing the representation explicitly. However, this strategy neglects a subtle yet critical trade-off inherent in DRL. A large representation rank can lead to an excessively complex representation function that can hinder the learning process (Goodfellow et al., 2016). Such complex models may not only demand a larger dataset for successful training but also exhibit high sensitivity to noise, impeding sample efficiency and overall performance.

Motivated by the relationship between cosine similarity and representation rank elucidated in Section 3.1, we propose an adaptive regularization technique. Our method uses the upper bounds derived in the preceding subsection to constrain the representation rank, conforming to the limitations stipulated by Equations (9) and (10). To this end, we introduce a hard constraint within the value function's optimization process. Formally,

$$\min_{\phi, w} \mathcal{L}_{VA}(\theta) = \frac{1}{2}(\phi^\top w - \mathbb{SG}\phi'^\top w)^2$$

$$\text{s.t. } \cos(\phi(s,a), \mathbb{SG}\overline{\phi(s',a')}) \leq (\|\phi(s,a)\|^2 + \gamma^2\|\overline{\phi(s',a')}\|^2 - \frac{\|r\|^2}{\|w\|^2})\frac{1}{2\gamma\|\phi(s,a)\|\|\overline{\phi(s',a')}\|},$$
(11)

where $\mathcal{L}_{VA}$ denotes standard value approximation loss and $\mathbb{SG}$ is the stopping gradient, reflecting the semi-gradient nature of the Bellman backup (Sutton & Barto, 2018). One can interpret that equation as adding an explicit desirable hard condition to the learning process of the value network. The introduced constraint effectively ensures that the representations satisfy conditions derived from the original Bellman equation, thereby aiding the value network's learning process. Applying the Lagrange multiplier technique to solve this optimization problem is challenging because the original problem is not convex. Hence, we cannot solve the problem using the primal-dual method like Haarnoja et al. (2018) does. Consequently, we utilize a penalty regularizer. We introduce the following regularizer into the value network learning process:

$$\mathcal{R}(\theta) = \text{ReLU}\Big( \cos(\phi(s,a), \mathbb{SG}\overline{\phi(s',a')}) -$$

$$\mathbb{SG}\big((\|\phi(s,a)\|^2 + \gamma^2\|\overline{\phi(s',a')}\|^2 - \frac{\|r\|^2}{\|w\|^2})\frac{1}{2\gamma\|\phi(s,a)\|\|\overline{\phi(s',a')}\|}\big)\Big),$$
(12)

where $\text{ReLU}(x) = \max\{0, x\}$. Incorporating this regularizing term offers generalized applicability across various DRL algorithms that utilize value approximation. Formally, the loss function can be rewritten

$$\mathcal{L}(\theta) = \mathcal{L}_{VA}(\theta) + \beta\mathcal{R}(\theta),$$
(13)

where $\beta$ serves as a hyper-parameter controlling the regularization effectiveness on the learning procedure. The working mechanism of the regularizer is as follows. When the input to the ReLU function is negative, it indicates the condition in Equation (9) is fullfilled. Under such circumstances, the regularizer ceases to influence the optimization trajectory, thereby allowing the original value approximation loss, $\mathcal{L}_{VA}$, to exclusively govern the optimization process. Conversely, when the ReLU input is positive, the regularizer contributes to optimizing the value function and minimizes the cosine similarity to ensure compliance with Equation (9). To empirically validate the utility of the proposed BEER regularizer, we integrate it with DQN (Mnih et al., 2015) for problems with discrete action spaces and with the Deterministic Policy Gradient (DPG) algorithm (Silver et al., 2014) for those involving continuous action spaces. We summarize BEER based on DPG in Algorithm 1.

### 3.3 AN ILLUSTRATIVE EXAMPLE

Does a very large representation rank harm the learning process of value function? To validate the adaptive regularization of the representation rank of BEER, we perform illustrative experiments on the Lunar Lander task. In this task, a spacecraft shall safely land on a designated landing pad. A policy with a return greater than 200 is considered optimal. We evaluate three algorithms: DQN, InFeR, and a relaxed version of BEER, which we integrate into the DQN framework without additional modifications. Comprehensive details of the experimental setup appear in Appendix C. The primary goal of a value network is a precise approximation of the optimal value function. Hence, we choose approximation error —a metric defined as the absolute difference between the estimated and the true value functions— as the principal criterion for model evaluation. This measure assesses the models' quality directly and precisely.

Figure 1 shows the results. Initially, we quantify the representation rank for each algorithm as

---

**Algorithm 1** BEER (based on DPG)

1: Init actor $\pi$, critic $Q$, targets $\pi'$, $Q'$ with random parameters $\theta, \varphi, \theta', \varphi'$, replay buffer $R$, noise distribution $\mathcal{N}(0, \sigma)$, regularization coefficient $\beta$, total timestep $T$
2: Init environment, get state $s$
3: **for** t = 1, T **do**
4: $\quad a_t = \pi_\theta(s_t) + \epsilon, \epsilon \sim \mathcal{N}$
5: $\quad$ Execute $a_t$, get $r_t, s_{t+1}$
6: $\quad$ Store $(s_t, a_t, r_t, s_{t+1})$ in $R$
7: $\quad$ Sample batch from $R$
8: $\quad$ Compute target value:
9: $\quad y_i = r_i + \gamma Q_{\varphi'}(s_{i+1}, \pi_{\theta'}(s_{i+1}))$
10: $\quad$ Update critic parameters $\varphi$ by minimizing $\mathcal{L}(\theta) = \frac{1}{N} \sum_i (y_i - Q_\varphi(s_i, a_i))^2 + \beta \mathcal{R}(\varphi)$ (Equation (13))
11: $\quad$ Update policy parameters $\theta$ using DPG
12: $\quad \nabla_\theta J \approx \frac{1}{N} \sum_i \nabla_a Q_\varphi(s_i, a_i) \nabla_\theta \pi_\theta(s_i)$
13: $\quad$ Update target network $\theta', \varphi'$
14: $\quad \theta' \leftarrow \tau\theta + (1-\tau)\theta', \varphi' \leftarrow \tau\varphi + (1-\tau)\varphi'$
15: **end for**

---

appears in Figure 1(b). We observe that BEER's representation rank is lower than those of both InFeR and DQN, thereby suggesting that the model learned with BEER is less complex. Then, we compute the approximation error. During the latter stage of the training process (0.9 to 1 ×40K time steps), BEER displays a substantially lower approximation error compared to both DQN and InFeR; that is, BEER generates a more accurate and reliable value function model. Figure 1(d) further confirms the superior performance of BEER. In brief, the experimental results verify that BEER improve the complexity of the model as originally envisioned by adaptively controlling the representation rank.

## 4 EXPERIMENTS

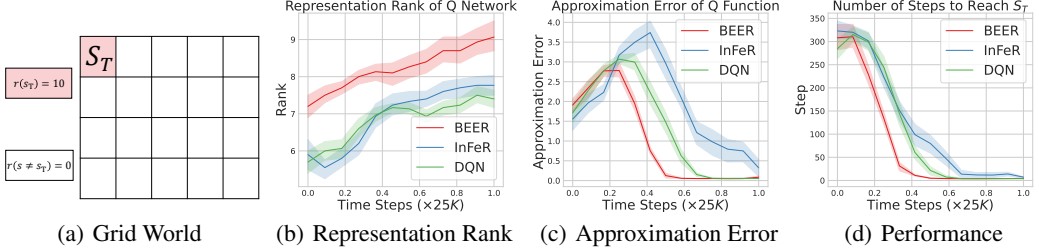

|(a) Grid World | (b) Representation Rank | (c) Approximation Error | (d) Performance |

Figure 2: Illustrative experiments on the Grid World task. We report the results over twenty random seeds. The shaded area represents a half standard deviation. (a) The grid world task. The initial state follows a uniform distribution over state space. The objective is to arrive at state $S_T$, which results in a reward (=10); Otherwise, it is zero. (b) Representation rank of tested algorithms. Our proposal, BEER, has the highest representation rank compared to InFeR and DQN. (c) Approximation error. The error of BEER is lower than that of InFeR and DQN. (d) The BEER algorithm requires the least time to reach $S_T$, i.e., it learns faster than the benchmarks.

In this section, we evaluate BEER concerning its regularization of representation rank and its empirical performance. Our evaluation focuses on three primary research questions. BEER algorithm tackles overly complex models with a potential high representation rank. Thus, one is interested in **i)** How does BEER perform in relatively straightforward tasks, such as the grid world environment? **ii)** Can BEER scale effectively to over-parameterized, high-complexity tasks? By over-parameterized tasks, we specifically refer to agents on DMControl tasks (Tunyasuvunakool et al., 2020), where the

agents are generally over-parameterized (Lutter et al., 2021; Kumar et al., 2021). **iii)** How does the approximation error of BEER compare to that of existing methods? A lower approximation error indicates a superior model in terms of complexity and performance. To ensure the reproducibility and fairness of our experiments (Henderson et al., 2018), all tests are based on ten random seeds. Besides, our BEER implementation does not incorporate any engineering trick that could potentially boost performance. In all experiments, we refrain from fine-tuning the hyperparameter $\beta$. We consistently use a value of $\beta = 1e-3$. We elaborate the experimental configurations in Appendix C to save space. The influence of the regularization coefficient on BEER's performance, as well as the experimental results of BEER on relatively simpler tasks, are presented in Appendices F and G, respectively. The impact of cosine similarity on rank representation is discussed in Appendix D. Whether the effects of the BEER regularizer are implicitly present in the value approximation algorithm is deliberated upon in Appendix E.

### 4.1 EFFECTIVENESS OF BEER ON SIMPLE ENVIRONMENTS

Here, we study the effectiveness of BEER in a simple environment with the grid world task as shown in Figure 2. As illustrated in Figure 2(b), BEER achieves a higher representation rank than the existing methods such as InFeR and DQN. Additionally, as shown in Figure 2(c), the approximation error of BEER is observably lower than other algorithms. Consequently, BEER also outperforms these methods concerning the number of steps required to reach the terminal state $S_T$ (see Figure 2(d)). Combined with the illustrative experiment in Section 3.3, these results demonstrate the adaptive regularization capability of BEER both on simple and complex tasks, which reflects our primary objective to design BEER.

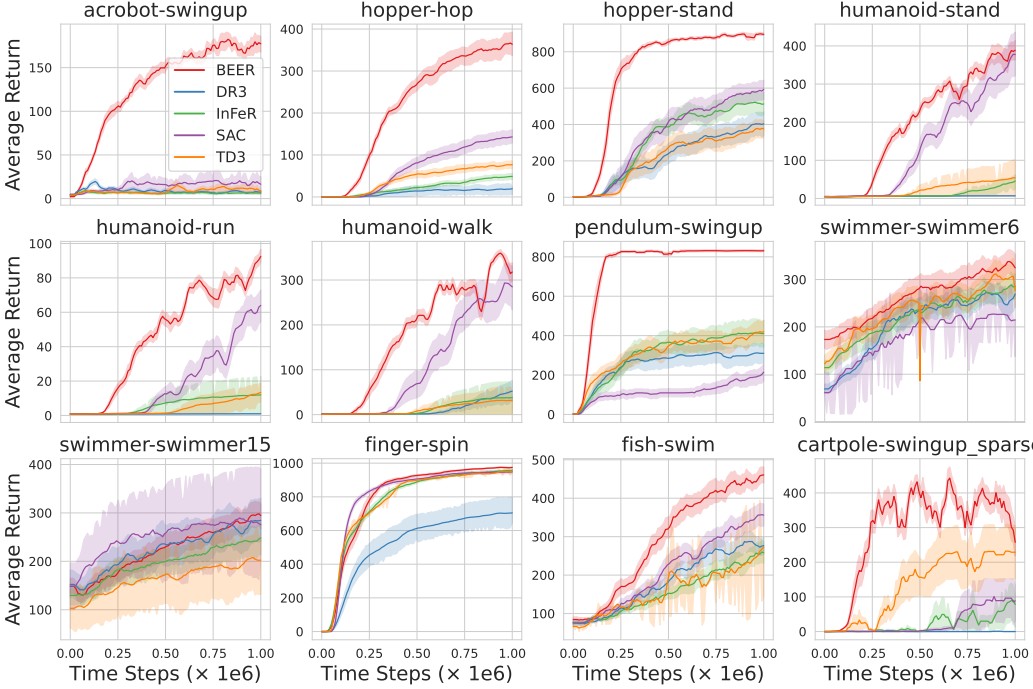

Figure 3: Performance curves for OpenAI gym continuous control tasks on DeepMind Control suite. The proposed algorithm, BEER, outperforms other tested algorithms significantly. The shaded region represents half of the standard deviation of the average evaluation over 10 seeds. The curves are smoothed with a moving average window of size ten.

### 4.2 SCALING UP TO COMPLEX CONTROL TASKS

DMControl (Tunyasuvunakool et al., 2020) serves as a standard benchmark suite for evaluating the capabilities of DRL agents in complex, continuous control tasks (Laskin et al., 2020b;a). We further

extend BEER's applicability to continuous control tasks and demonstrate the performance of the BEER regularizer on the 12 challenging (Yarats & Kostrikov, 2020) DMControl tasks, where we combine BEER with the deterministic policy gradient method (Silver et al., 2014; Lillicrap et al., 2016). Our primary objective in this study is to investigate the BEER's efficacy concerning automatic regularization of representation rank. We select the following baseline algorithm: i) DR3 (Kumar et al., 2021), which unboundedly maximizes representation rank by maximizing the inner product of two adjacent representations of the Q network; ii) InFeR, which consistently maximizes the rank with the help of designed auxiliary task; iii) TD3 (Fujimoto et al., 2018) and iv) SAC, both of which have already shown robust performance a spectrum of tasks. The performance curve, illustrated in Figure 3, demonstrates the sample efficiency and superior performance of BEER. We report the best average scores over the 1 million time steps in Table 1, where the best performance of BEER outperforms the other tested algorithm by a large margin. These results demonstrate the sample efficiency and performance of BEER.

Table 1: 10-Seed peak scores over 1M Steps on DMC. BEER demonstrates the best performance on the majority (**12** out of **12**) tasks by a large margin. Specifically, the BEER algorithm outperforms DR3, InFeR, SAC, and TD3 by 140%, 96.8%, 60%, and 101.9%, respectively. The best score is marked with the blue color box. ± corresponds to a standard deviation over ten trials.

| Domain | Task | BEER | DR3 | InFeR | SAC | TD3 |
|--------|------|------|-----|-------|-----|-----|
| Acrobot | Swingup | 260.5 ± 42.9 | 46.2 ± 14.0 | 18.9 ± 13.6 | 43.2 ± 63.5 | 32.3 ± 27.4 |
| Hopper | Hop | 383.6 ± 115.8 | 22.0 ± 50.0 | 58.1 ± 37.6 | 149.0 ± 73.6 | 91.2 ± 46.9 |
| Hopper | Stand | 929.3 ± 29.7 | 465.6 ± 276.3 | 563.4 ± 256.5 | 650.5 ± 238.5 | 443.0 ± 208.6 |
| Humanoid | Stand | 471.3 ± 92.5 | 7.4 ± 1.0 | 50.8 ± 87.9 | 418.1 ± 278.7 | 58.0 ± 151.6 |
| Humanoid | Run | 107.1 ± 10.6 | 1.1 ± 0.2 | 12.4 ± 32.9 | 78.7 ± 43.4 | 15.1 ± 29.5 |
| Humanoid | Walk | 393.2 ± 38.2 | 55.8 ± 107.2 | 39.0 ± 108.9 | 329.0 ± 202.0 | 33.5 ± 92.0 |
| Pendulum | Swingup | 833.2 ± 22.0 | 331.9 ± 228.8 | 456.4 ± 315.5 | 270.7 ± 228.1 | 453.8 ± 241.9 |
| Swimmer | Swimmer6 | 398.0 ± 123.8 | 321.1 ± 100.4 | 332.8 ± 125.3 | 243.8 ± 74.4 | 321.9 ± 148.4 |
| Swimmer | Swimmer15 | 345.5 ± 110.2 | 320.6 ± 165.6 | 283.8 ± 155.4 | 314.1 ± 198.5 | 226.7 ± 177.9 |
| Finger | Spin | 983.6 ± 6.8 | 715.2 ± 387.5 | 966.0 ± 21.8 | 956.5 ± 43.0 | 957.9 ± 26.9 |
| Fish | Swim | 573.2 ± 103.4 | 377.5 ± 123.4 | 335.7 ± 133.9 | 418.2 ± 127.1 | 316.0 ± 124.6 |
| Cartpole | SwingupSparse | 750.8 ± 61.8 | 15.0 ± 39.1 | 148.8 ± 235.2 | 147.6 ± 295.5 | 235.0 ± 356.9 |
| Average | Score | 535.8 | 223.3 | 272.2 | 335.0 | 265.4 |

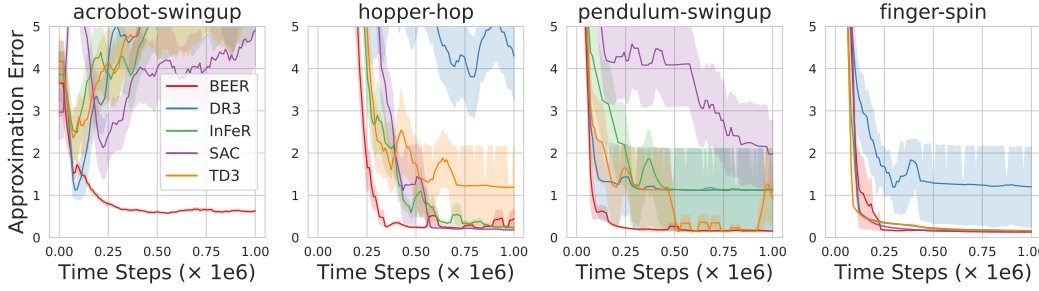

Figure 4: Approximation error curves. The results demonstrate that the approximation error of BEER is empirically minimal when compared to other algorithms.

### 4.3 BEER REDUCES APPROXIMATION ERROR.

The results discussed above establish the superiority of BEER that stems from the adaptive regularization of representation rank. We continue to validate the quality of the learned models in terms of approximation error. We select four representative tasks to measure the models, where BEER has a better performance on three and a matching performance on one of the tasks (finger-spin). As demonstrated in Figure 4, BEER maintains a lower approximation error than the baselines. The results are consistent with our previous theoretical analysis, thus validating its utility.

## 5 RELATED WORK

In this section, we dive into the intersection of RL with NNs and the growing emphasis on NNs-centric DRL. The differences between our work and previous studies are also discussed in this section.

### 5.1 RL WITH NNS

Prior to the era of DL, function approximators in RL are either tabular (Sutton & Barto, 2018) or utilized learnable weights based on handcrafted features. (Barreto et al., 2017). DQN (Mnih et al., 2015), which stimulates the tremendous development of DRL, leverages NNs to approximate Q functions. The subsequent work (Schulman et al., 2017; Lillicrap et al., 2016; Fujimoto et al., 2018; Haarnoja et al., 2018; He & Hou, 2020), which focuses on the RL side, utilizes NNs as general function approximators. As representation learning techniques (Devlin et al., 2019; Oord et al., 2018; He et al., 2020; Chen et al., 2020a; He et al., 2022) continue to evolve, an increasing number of these methods are being incorporated into the field of RL (Jaderberg et al., 2016; Guo et al., 2020; Ghosh & Bellemare, 2020; Laskin et al., 2020b; Yarats et al., 2021a;b; Laskin et al., 2020a; Oord et al., 2018), because the learned representations by NNs with these methods are beneficial for RL agents and also the downstream tasks. Our work is distinct because it considers the properties that RL should possess within the context of DL settings. Unlike some approaches that predominantly draw upon intuitions from the field of DL, our work aims for an integrated understanding of both domains.

### 5.2 NNS-CENTRIC DRL

Some studies focus on leveraging NNs to enhance the capabilities of DRL. Examples include approaches that employ NNs to learn multiple Q-values (Anschel et al., 2017; Lan et al., 2019; Chen et al., 2020b), simultaneously learn state value and advantage value (Anschel et al., 2017), and deep successor features (Barreto et al., 2017). Researchers have gradually recognized that the integration of NNs with RL enables specific characteristics, such as the distinguishable representation property (He et al., 2023a) and the domination of top-subspaces in the learning process (Lyle et al., 2021; He et al., 2023b). However, this integration also introduces particular challenges that hinder DRL, such as the catastrophic forgetting (Khetarpal et al., 2022), the capacity loss (Lyle et al., 2022), the plasticity loss (Lyle et al., 2023), dormant neurons (Sokar et al., 2023), primacy bias (Nikishin et al., 2022), and etc. These studies aim to identify and rectify NNs' attributes that are detrimental to RL. For instance, (Lyle et al., 2022; 2023) assert that NNs become less adaptive to new information as learning progresses, thereby experiencing a capacity/plasticity loss that can be alleviated by increasing the neural network's representation rank. Recent literature discusses how to maximize representation rank in bandit settings and RL settings (Papini et al., 2021b;a; Zhang et al., 2023), which cannot inherently bound the cosine similarity in this work.

In contrast, our work diverges from these studies by originating from the inherent properties of RL rather than focusing solely on the NNs component. We establish a connection with representation rank and introduce a novel method for adaptively controlling it, providing a fundamentally different starting point compared to previous research. In terms of the form of the regularizer, both previous works Kumar et al. (2021); He et al. (2023a) and ours involve the inner product form of the representation. Nevertheless, both Kumar et al. (2021) and He et al. (2023a) utilize an unboundedly maximizing form. In contrast, our work starts with the Bellman equation and introduces an adaptive regularizer that will not unboundedly maximize the representation rank.

## 6 CONCLUSION

In this study, we discussed a significant issue of representation rank in DRL, which the current literature neglects to a great extent: How to adaptively control the representation rank of DRL agents. By rigorously analyzing an inherent property of the Bellman equation in the DRL setting, we introduced the theoretically grounded regularizer BEER, a novel approach to adaptively control the representation rank. Our theoretical and empirical analyses demonstrated the effectiveness of BEER, outperforming the existing methods that focus on the representation rank. In our future works, we plan to validate the efficacy of BEER on other benchmarks. Our work opens new avenues for understanding the role of representation rank in DRL and offers an adaptively practical tool to control the rank and thus improve agent performance.

ACKNOWLEDGMENTS

The authors thank anonymous reviewers, the area chair, and the senior area chair for fair evaluations and professional work. This research was supported by Grant 01IS20051 and Grant 16KISK035 from the German Federal Ministry of Education and Research (BMBF).

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

## A    PROOF OF THEOREM 1

**Theorem 1.** *Under Assumption 1, given Q value $Q(s,a) = \phi(s,a)^\top w$,*

$$\langle \phi(s,a), \overline{\phi(s',a')} \rangle \leq (\|\phi(s,a)\|^2 + \gamma^2 \|\overline{\phi(s',a')}\|^2 - \frac{\|r\|^2}{\|w\|^2}) \frac{1}{2\gamma}, \tag{9}$$

*where $\langle \cdot, \cdot \rangle$ represents the inner product, $\phi$ is a state-action representation vector, $w$ is a weight, and $\overline{\phi(s',a')} = \mathbb{E}_{s',a'} \phi(s',a')$ denotes the expectation of the representation of the next state action pair.*

*Proof.* We start with the Bellman equation for Q values, i.e.,

$$Q(s,a) = r(s,a) + \gamma \mathbb{E}_{s' \sim p(s'|s,a), a' \sim \pi(s')}[Q(s',a')]. \tag{14}$$

Substituting the representation $Q(s,a) = \phi(s,a)^\top w$ into the equation, we get

$$\phi(s,a)^\top w = r + \gamma \overline{\phi(s',a')}^\top w]. \tag{15}$$

Taking the norm of both sides gives us

$$\phi(s,a)^\top w - \gamma \overline{\phi(s',a')}^\top w = r, \tag{16}$$

$$\|(\phi(s,a)^\top - \gamma \overline{\phi(s',a')}^\top)w\| = \|r\|, \tag{17}$$

$$\|\phi(s,a)^\top - \gamma \overline{\phi(s',a')}^\top\| \|w\| \geq \|r\|, \tag{18}$$

$$\|\phi(s,a)^\top - \gamma \overline{\phi(s',a')}^\top\| \geq \frac{\|r\|}{\|w\|}, \tag{19}$$

where the third line follows by the Cauchy-Schwarz inequality. Taking the square of both sides, we obtain

$$\|\phi(s,a) - \gamma \overline{\phi(s',a')}\|^2 \geq \frac{\|r\|^2}{\|w\|^2}. \tag{20}$$

Expanding the square of the norm, we find

$$\|\phi(s,a) - \gamma \overline{\phi(s',a')}\|^2 \tag{21}$$

$$= \langle \phi(s,a) - \gamma \overline{\phi(s',a')}, \phi(s,a) - \gamma \overline{\phi(s',a')} \rangle \tag{22}$$

$$= \|\phi(s,a)\|^2 - 2\gamma \langle \phi(s,a), \overline{\phi(s',a')} \rangle + \gamma^2 \|\overline{\phi(s',a')}\|^2 \geq \frac{\|r\|^2}{\|w\|^2} \tag{23}$$

Finally, rearranging the terms yields

$$\langle \phi(s,a), \overline{\phi(s',a')} \rangle \leq (\|\phi(s,a)\|^2 + \gamma^2 \|\overline{\phi(s',a')}\|^2 - \frac{\|r\|^2}{\|w\|^2}) \frac{1}{2\gamma}, \tag{24}$$

which concludes the proof.  □

## B    SIMILARITY UPPER BOUND UNDER THE BELLMAN OPTIMALITY EQUATION

We discuss the similarity upper bound under the Bellman optimal equation. That induces another BEER regularizer, which is slightly different from Equation (12). We first introduce an upper bound on the representations of two consecutive representations based on the Bellman optimality Equation. We then derive the related bound on the cosine similarity. Afterward, we introduce a BEER regularizer based on the bound combined with the DQN algorithm. Such a regularizer is used in Figures 1 and 2.

**Theorem 2.** *Under Assumption 1, given Q value $Q(s,a) = \phi(s,a)^\top w$,*

$$\langle \phi(s,a), \widehat{\phi(s',a')} \rangle \leq (\|\phi(s,a)\|^2 + \gamma^2 \|\widehat{\phi(s',a')}\|^2 - \frac{\|r\|^2}{\|w\|^2}) \frac{1}{2\gamma}, \tag{25}$$

*where $\langle \cdot, \cdot \rangle$ represents the inner product, $\phi$ is a state-action representation vector, $w$ is a weight. And $\widehat{\phi(s',a')} = \mathbb{E}_{s' \sim p(s'|s,a)} \phi(s',a')$ with $a' = \arg\max_{a' \in \mathcal{A}} Q(s',a')$.*

The proof is similar to Appendix A.

*Proof.* We begin with the Bellman optimality equation for Q values, i.e.,

$$Q(s,a) = r(s,a) + \gamma \mathbb{E}_{s' \sim p(s'|s,a)} [\max_{a' \in \mathcal{A}} Q(s',a')] \tag{26}$$

Substituting the representation $Q(s,a) = \phi(s,a)^\top w$ into the equation, we get

$$\phi(s,a)^\top w = r(s,a) + \gamma \widehat{\phi(s',a')}^\top w \tag{27}$$

Taking the norm of both sides yields

$$\phi(s,a)^\top w - \gamma \widehat{\phi(s',a')}^\top w = r, \tag{28}$$

$$\|(\phi(s,a)^\top - \gamma \widehat{\phi(s',a')}^\top)w\| = \|r\|, \tag{29}$$

$$\|\phi(s,a)^\top - \gamma \widehat{\phi(s',a')}^\top \| \|w\| \geq \|r\|, \tag{30}$$

$$\|\phi(s,a)^\top - \gamma \widehat{\phi(s',a')}^\top \| \geq \frac{\|r\|}{\|w\|}, \tag{31}$$

where the third line follows from the Cauchy-Schwarz inequality. Squaring both sides, we find

$$\|\phi(s,a) - \gamma \widehat{\phi(s',a')}^\top \|^2 \geq \frac{\|r\|^2}{\|w\|^2}. \tag{32}$$

Expanding the square of the norm, we find

$$\|\phi(s,a) - \gamma \widehat{\phi(s',a')}\|^2 \tag{33}$$

$$= \langle \phi(s,a) - \gamma \widehat{\phi(s',a')}, \phi(s,a) - \gamma \widehat{\phi(s',a')} \rangle \tag{34}$$

$$= \|\phi(s,a)\|^2 - 2\gamma \langle \phi(s,a), \widehat{\phi(s',a')} \rangle + \gamma^2 \|\widehat{\phi(s',a')}\|^2 \geq \frac{\|r\|^2}{\|w\|^2} \tag{35}$$

Finally, we obtain the inequality stated in equation 25:

$$\langle \phi(s,a), \widehat{\phi(s',a')} \rangle \leq (\|\phi(s,a)\|^2 + \gamma^2 \|\widehat{\phi(s',a')}\|^2 - \frac{\|r\|^2}{\|w\|^2}) \frac{1}{2\gamma}, \tag{36}$$

which completes the proof. □

The following remark arises from Theorem 2.
*Remark 2.* Under Assumption 1, given Q value $Q(s,a) = \phi(s,a)^\top w$, then

$$\cos(\phi(s,a), \widehat{\phi(s',a')}) \leq (\|\phi(s,a)\|^2 + \gamma^2 \|\widehat{\phi(s',a')}\|^2 - \frac{\|r\|^2}{\|w\|^2}) \frac{1}{2\gamma \|\phi(s,a)\| \|\widehat{\phi(s',a')}\|}. \tag{37}$$

According to Theorem 2 and Remark 2, we obtain the following regularizer:

$$\mathcal{R}(\theta) = \text{ReLU}\Big( \cos(\phi(s,a), \mathbb{SG}\widehat{\phi(s',a')}) - $$
$$(\|\phi(s,a)\|^2 + \gamma^2 \|\widehat{\phi(s',a')}\|^2 - \frac{\|r\|^2}{\|w\|^2}) \frac{1}{2\gamma \|\phi(s,a)\| \|\widehat{\phi(s',a')}\|} \Big), \tag{38}$$

which is used in Figures 1 and 2.

# C  ADDITIONAL DETAILS REGARDING EXPERIMENTS

In this section, we intend to furnish additional insights and specifications concerning the experimental setups employed to derive the results presented in the main manuscript. This systematic presentation of our experimental procedures aims to promote transparency, facilitate replication, and enable future research to build upon these foundational insights.

**Algorithm Design.** Our goal is to analyze the properties of the BEER regularizer. As we previously claimed, it can be integrated with DRL algorithms that involve value approximation. In addition to combining it with DQN ( Figures 1 and 2), we also integrate the BEER algorithm with DPG (Silver et al., 2014), as listed in Algorithm 1. The purpose of this integration is to focus on studying the intrinsic properties of BEER itself, thereby demonstrating the superiority of the BEER regularizer. DPG propagates gradients through the value network when optimizing the policy network. A high-quality value network leads to a high-quality policy network. Our BEER regularizer is primarily designed to achieve a better value network by adaptively controlling the representation rank. Therefore, we chose to combine the BEER regularizer with DPG. In our implementation of BEER based on DPG, we exclusively use pure DPG. The exploration scheme of BEER+DPG involves adding Gaussian noise to the policy network, as done in the TD3 paper (Fujimoto et al., 2018). It's worth noting that, for a more accurate measurement of the BEER algorithm, we have not included any additional performance-enhancing techniques such as double Q clipping or target noise smoothing for the BEER algorithm based on DPG. Comparing BEER based on DPG with TD3 serves as a natural **ablation** experiment. This is because BEER is combined only with DPG without any other tricks, while TD3 has already been widely proven to be one of the best DPG algorithms across a broad range of scenarios (Yarats et al., 2021b; Fujimoto et al., 2018; Haarnoja et al., 2018).

**Implementation Consistency.** In our quest to maintain scientific rigor, we have imposed deterministic controls by fixing all random seeds, encompassing libraries such as PyTorch, Numpy, Gym, Random, and CUDA. Unless otherwise noted, all algorithms under study are assessed across 10 predetermined random seeds. Our code is available at https://github.com/sweetice/BEER-ICLR2024.

**Computational resources.** We conducted all experiments on a single GPU server. This server is equipped with 8 GeForce 2080 Ti GPUs and has 70 CPU logical cores, allowing for the parallel execution of 70 trials. Computational tasks for 70 runs can be completed within eight hours.

**Figure 1.** We conducted the experiments concerning this figure on the Lunar Lander-1 environment. We calculated the ground truth Q-values through Monte Carlo methods Fujimoto et al. (2018). The error bars in the figures represent half of the standard deviation across multiple trials. Unless specified otherwise, such error bars in all subsequent figures denote half of a standard deviation across 10 unique seeds. The implementation of the three algorithms is nearly identical, except for the portions unique to each algorithm. For specific hyperparameters, please refer to Table 2.

**Figure 2.** The grid world is shown in Figure 2(a). If the agent arrives at $S_T$, it gets a reward of 10, and other states get a reward of 0. We present the remaining hyper-parameters for the grid world in Table 3.

**Figure 3.** Note that we select 12 challenging tasks from dmcontrol (Tunyasuvunakool et al., 2020). Our selection is based on the data reported by Yarats & Kostrikov (2020), where we opt for the twelve tasks with the highest level of challenge to present in the main text. Results for simpler tasks are also shown in Table 6. For Figure 3, we examine the algorithmic performance on continuous control tasks in the DeepMind Control Suite. The interaction protocol employs the gym Python library (Brockman et al., 2016). The shaded regions depict half a standard deviation around the mean evaluation score from 10 different seeds. Additionally, the performance curves are smoothened using a moving average window of 10. Evaluations were conducted over 1 million timesteps, with average returns recorded at intervals of 10,000 timesteps over 10 episodes. We present the remaining hyper-parameters for the DMControl in Table 4.

**Figure 4.** The approximation error is normalized relative to the true Q value; i.e., approximation error $= \frac{\text{orignal approximation error}}{\text{True Q}+\epsilon}$, where $\epsilon = 1e-6$ is a small number to stabilize the computational process. The purpose of normalization is to mitigate the impact of the value scale on the approximation error, which arises due to performance disparities among various algorithms so that we can compare the approximation error of different algorithms. The performance of the

algorithm under test is evaluated across 1 million timesteps. We assess the approximation error at intervals of 10,000 timesteps. For clarity in visualization, the y-axis is limited to the range $[0, 5]$.

**Table 1.** First, we calculate the average return obtained by the algorithm across ten episodes within one million steps. We then select the best average value from this set. Next, we choose ten different random seeds and calculate similar values for each seed. The final data is obtained by averaging these ten values.

Table 2: Hyper-parameters settings for lunar lander-v1 task (Figure 1). For InFeR, we use its default hyper-parameters. For every $5e3$ step, we evaluate the algorithms.

| Hyper-parameter | Value |
|---|:---:|
| *Shared hyper-parameters* | |
| Dimension of state space | 8 |
| Action space | Discrete(4): up, down, left, right |
| Discount ($\gamma$) | 0.99 |
| Replay buffer size | $10^5$ |
| Optimizer | Adam (Kingma & Ba, 2015) |
| Learning rate for Q-network | $1 \times 1e^{-3}$ |
| Number of hidden layers for all networks | 3 |
| Number of hidden units per layer | 64 |
| Activation function | ReLU |
| Mini-batch size | 64 |
| Random starting exploration time steps | $10^3$ |
| Target smoothing coefficient ($\eta$) | 0.005 |
| Gradient clipping | False |
| Exploration method | Epsilon-Greedy |
| $\epsilon$ (Exploration) | 0.1 |
| Evaluation episode | 10 |
| Number of steps | $4e4$ |
| $\epsilon$ for rank computing | 0.01 |
| *BEER* | |
| BEER coefficient ($\beta$) | $5e-3$ |
| *InFeR* | |
| Number of head (k) | 10 |
| $\beta$ | 100 |
| $\alpha$ | 0.1 |

Table 3: Hyper-parameters settings for Grid World experiments. For every $5e3$ step, we evaluate the algorithms.

| Hyper-parameter | Value |
|---|:---:|
| *Shared hyper-parameters* | |
| State space | integer: from 0 to 19 |
| Action space | Discrete(4): up, down, left, right |
| Discount ($\gamma$) | 0.99 |
| Replay buffer size | $10^5$ |
| Optimizer | Adam (Kingma & Ba, 2015) |
| Learning rate for Q-network | $1 \times 10^{-4}$ |
| Number of hidden layers for all networks | 2 |
| Number of hidden units per layer | 32 |
| Activation function | ReLU |
| Mini-batch size | 64 |
| Random starting exploration time steps | $10^3$ |
| Target smoothing coefficient ($\eta$) | 0.005 |
| Gradient clipping | False |
| Exploration Method | Epsilon-Greedy |
| $\epsilon$ (Exploration) | 0.1 |
| Evaluation episode | 10 |
| Number of timesteps | $2.5e4$ |
| $\epsilon$ for rank computing | 0.01 |
| *BEER* | |
| BEER coefficient ($\beta$) | $5e-3$ |

Table 4: Hyper-parameters settings for DMcontrol experiments (Figures 3 and 4 and Table 1).

| Hyper-parameter | Value |
|---|:---:|
| *Shared hyper-parameters* | |
| Discount ($\gamma$) | 0.99 |
| Replay buffer size | $10^6$ |
| Optimizer | Adam (Kingma & Ba, 2015) |
| Learning rate for actor | $3 \times 10^{-4}$ |
| Learning rate for critic | $3 \times 10^{-4}$ |
| Number of hidden layers for all networks | 2 |
| Number of hidden units per layer | 256 |
| Activation function | ReLU |
| Mini-batch size | 256 |
| Random starting exploration time steps | $2.5 \times 10^4$ |
| Target smoothing coefficient ($\eta$) | 0.005 |
| Gradient clipping | False |
| $\epsilon$ for rank computing | 0.01 |
| *TD3* | |
| Variance of exploration noise | 0.2 |
| Variance of target policy smoothing | 0.2 |
| Noise clip range | $[-0.5, 0.5]$ |
| Delayed policy update frequency | 2 |
| Target update interval ($d$) | 2 |
| *SAC* | |
| Target entropy | - dim of $\mathcal{A}$ |
| Learning rate for $\alpha$ | $1 \times 10^{-4}$ |
| Target update interval ($d$) | 2 |
| *InFeR* | |
| Number of head (k) | 10 |
| $\beta$ | 100 |
| $\alpha$ | 0.1 |
| *DR3* | |
| Regularization coefficient ($c_0$) | 5e-3 |
| *BEER* | |
| BEER coefficient ($\beta$) | $1 \times 10^{-3}$ |
| Variance of exploration noise | 0.2 |

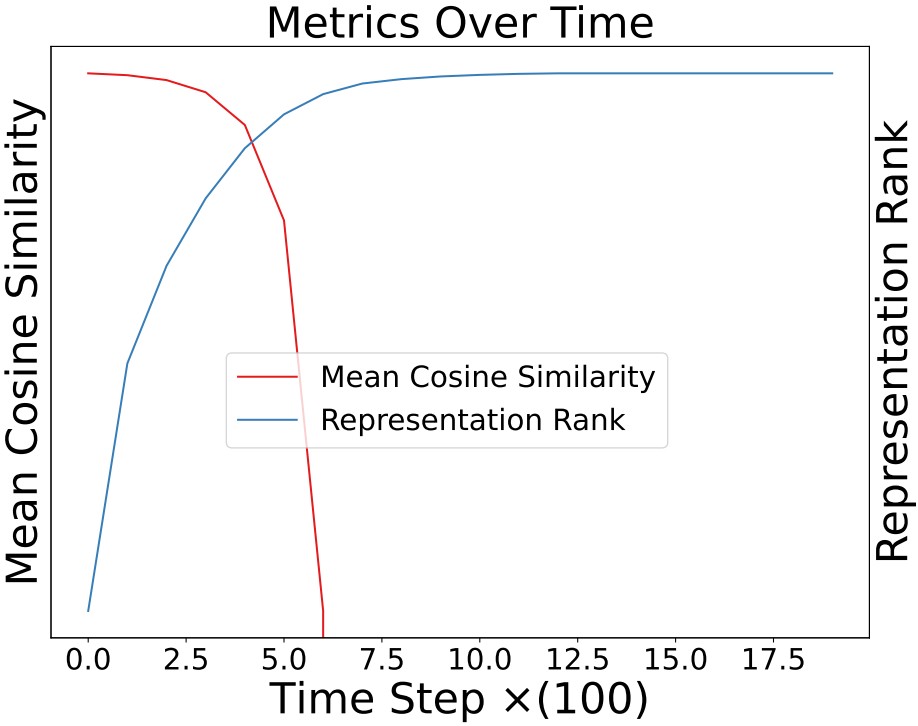

Figure 5: Relationship between cosine similarity and representation rank when minimizing cosine similarity. Our experiment demonstrates the effective control of cosine similarity for representation rank.

## D EMPIRICAL VALIDATION OF COSINE SIMILARITY'S CONTROL ON REPRESENTATION RANK

In Section 3.1, we discussed how cosine similarity can control representation rank. Here, we empirically validate the control effect of cosine similarity on representation rank to gain an intuitive understanding. The results of the experiment are shown in Figure 5. In this experiment, we first generate a matrix with very high cosine similarity, resulting in a very low representation rank. We randomly sample a mini-batch of row vectors from this matrix and minimize the cosine similarity between these vectors. We then track the relationship between cosine similarity and the representation rank we've defined, as displayed in Figure 5. For better visual clarity, we use a logarithmic scale on the y-axis to better observe the relative change trends between these two metrics. We find that a slight reduction in cosine similarity immediately leads to an increase in representation rank, demonstrating the effective control of cosine similarity for the representation rank.

Specifically, we first generate a $256 \times 256$ dimensional matrix and use the Adam optimizer with a learning rate of 5e-3. We sample 64 row vectors from the matrix each time. Every 100 training steps, we record the cosine similarity among the vectors in the matrix as well as the representation rank, using an epsilon value of 0.05 for calculating the latter. The experiment is conducted over 2,000 steps. In the later stages of the experiment, the cosine similarity becomes too small to be within the display range. At this point, the representation rank is almost full rank.

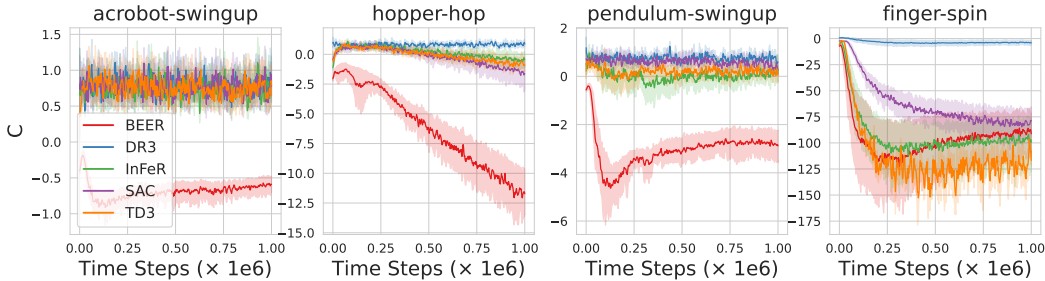

Figure 6: The results demonstrate that the tested algorithms don't hold the bound across all the tasks. However, BEER makes the agent hold the bound. Combined with the performance results (see Figure 3), these demonstrate that BEER effectively ensures the satisfaction of this bound, thereby adaptively regularizing the rank and ultimately improving performance.

## E IS THE BEER REGULARIZER IMPLICITLY INCORPORATED INTO VALUE APPROXIMATION ALGORITHMS?

Our theory and regularizers are built upon the Bellman equation. Almost all the value-based DRL algorithms leverage the Bellman equation. Thus, one would suspect that this regularizer effect would also be implicitly satisfied by these algorithms without BEER since these algorithms also use the Bellman equation. Kumar et al. (2021) shows that the dynamics of NNs result in feature co-adaptation. Such co-adaptation makes the inner product of representations grow fast, which potentially makes DRL agents fail to hold the bound. To corroborate this claim, we conduct additional experiments. During the training process, we collect data every two thousand training steps to ascertain whether the bound Equation (9) is preserved or not.

We define

$$C = \langle \phi(s,a), \overline{\phi}(s',a') \rangle - (\|\phi(s,a)\|^2 + \gamma^2 \|\overline{\phi}(s',a')\|^2 - \frac{\|r\|^2}{\|w\|^2}) \frac{1}{2\gamma}. \tag{39}$$

$C <= 0$ means the bound in Equation (9) holds, otherwise it does not. The results of the experiments are presented in Figure 6. The results demonstrate that the tested algorithms don't consistently hold the bound. While BEER makes the agent hold the bound.

Specifically, BEER maintains this bound across the first three tasks, thereby significantly outperforming other algorithms on these tasks (see Figure 3). In contrast, alternative algorithms under test, even those (DR3 and InFeR) specifically designed to increase the representation rank, fail to maintain this bound, resulting in inferior performance. On the finger-spin task, all algorithms meet this bound, and there is no discernible difference in performance among them. This demonstrates that our designed algorithm effectively ensures the satisfaction of this bound, thereby adaptively regularizing the rank and ultimately improving performance. These experimental findings are consistent with our theoretical analysis.

## F    THE COEFFICIENT OF BEER

In this section, we investigate the selection of hyper-parameter $\beta$. We continue to use the four environments identified in the main text as representative examples and consider eight different values for the hyperparameter $\beta$. Our experimental results indicate that starting from 0.001, as the hyperparameter increases, the algorithm's performance gradually deteriorates due to the increasing interference of the regularization effect on the normal learning of the value function. The results suggest that a $\beta$ value range less than or equal to 0.001 is reasonable. Performance differences at lower $\beta$ values are not significant. Therefore, in the main text, we consistently use $1e-3$ as the hyperparameter.

Table 5: 10-Seed peak scores over 1M Steps on DMC. The best score is marked with the blue color box. The results suggest that a $\beta$ value range less than or equal to 0.001 is reasonable. Performance differences at lower $\beta$ values are not significant. Therefore, in the main text, we consistently use $1e-3$ as the hyperparameter.

| **Domain** | **Task** | **0.0001** | **0.0005** | **0.001** | **0.005** | **0.01** | **0.1** | **1.0** | **10.0** |
|---|---|---|---|---|---|---|---|---|---|
| Acrobot | Swingup | 232.8 | 191.7 | 243.3 | 171.9 | 171.8 | 148.2 | 148.9 | 81.1 |
| Hopper | Hop | 376.1 | 404.5 | 365.7 | 282.3 | 362.7 | 217.9 | 0.4 | 0.6 |
| Pendulum | Swingup | 830.0 | 829.8 | 830.0 | 829.1 | 828.8 | 616.6 | 660.2 | 25.5 |
| Finger | Spin | 974.8 | 978.2 | 979.2 | 975.1 | 981.7 | 764.4 | 384.1 | 1.3 |
| Average | Score | 603.4 | 601.0 | 604.6 | 564.6 | 586.2 | 436.8 | 298.4 | 27.1 |

## G    ADDITIONAL EXPERIMENTAL RESULTS ON DMCONTROL SUITE

In Section 4, we report BEER performance in 12 challenging tasks. In Table 6, we report the performance of the tested algorithms on easier DMC tasks. The results show that BEER has a slight performance advantage over the other algorithms, although not as large as the advantage on the other challenging tasks. This is because these tasks are relatively simple according to the results reported by Yarats & Kostrikov (2020). So the performance difference between the algorithms is not too large. Less capable algorithms are also able to perform very well.

Table 6: 10-Seed peak scores over 1M Steps on 9 easy DMC tasks. On these tasks, BEER has a slight performance advantage over the other algorithms, although not as large as the advantage on the other challenging tasks. This is because these tasks are relatively simple, so the performance difference between the algorithms is not too large. Less capable algorithms are also able to perform very well. $\pm$ corresponds to a standard deviation over ten trials.

| Domain | Task | BEER | DR3 | InFeR | SAC | TD3 |
|---|---|---|---|---|---|---|
| Cartpole | Swingup | $879.2 \pm 1.7$ | $875.8 \pm 6.9$ | $875.1 \pm 7.4$ | $869.5 \pm 5.7$ | $872.3 \pm 7.4$ |
| Cheetah | Run | $883.4 \pm 12.2$ | $863.9 \pm 23.7$ | $857.3 \pm 37.7$ | $874.5 \pm 19.8$ | $853.4 \pm 37.6$ |
| Finger | TurnEasy | $976.5 \pm 6.5$ | $961.2 \pm 50.5$ | $902.1 \pm 89.3$ | $949.2 \pm 47.8$ | $975.6 \pm 12.2$ |
| Finger | TurnHard | $949.2 \pm 37.1$ | $796.8 \pm 123.6$ | $816.9 \pm 158.4$ | $896.5 \pm 97.6$ | $967.9 \pm 10.0$ |
| PointMass | Easy | $893.7 \pm 4.7$ | $815.3 \pm 270.7$ | $903.0 \pm 12.3$ | $902.3 \pm 12.5$ | $903.1 \pm 12.3$ |
| Reacher | Easy | $985.4 \pm 3.6$ | $982.5 \pm 3.9$ | $984.2 \pm 2.8$ | $984.5 \pm 4.0$ | $984.2 \pm 3.5$ |
| Reacher | Hard | $975.4 \pm 2.0$ | $976.9 \pm 1.2$ | $976.4 \pm 5.2$ | $980.7 \pm 0.9$ | $976.6 \pm 0.1$ |
| Walker | Stand | $983.6 \pm 3.9$ | $988.0 \pm 3.8$ | $987.2 \pm 2.5$ | $988.2 \pm 1.9$ | $985.3 \pm 2.7$ |
| Walker | Run | $793.6 \pm 12.6$ | $653.7 \pm 129.5$ | $697.7 \pm 61.8$ | $791.0 \pm 26.1$ | $680.8 \pm 108.4$ |
| Average | Score | $924.4$ | $879.3$ | $888.9$ | $915.2$ | $911.0$ |

## H    COMPUTATIONAL OVERHEAD OF BEER

In this section, we assess the computational impact of our proposed method, BEER, compared to other algorithms. Our evaluation focuses on two main aspects: the number of learnable parameters and the runtime analysis.

**Learnable Parameters.** Unlike methods such as InFeR, which introduce multiple additional heads to the value networks, thereby increasing the number of learnable parameters, BEER functions as a regularizer without adding any new learnable parameters to the base algorithm. This is a significant advantage as it maintains computational efficiency.

**Runtime Analysis.** To evaluate the runtime efficiency of BEER, we conduct comparative experiments against algorithms such as InFeR, TD3, SAC, and DR3. These experiments were performed on a GPU server equipped with an Intel(R) Xeon(R) Gold 6240 CPU @ 2.60GHz and NVIDIA GeForce RTX 2080 Ti GPUs. Each task is executed sequentially under the same evaluation protocol. Table 7 presents the runtime in hours for each algorithm across various environments. As evident from Table Table 7, BEER demonstrates competitive runtime efficiency, further highlighting its computational viability for practical applications in deep reinforcement learning.

Table 7: Runtime comparison of BEER and other algorithms.

| Environment | BEER | INFER | TD3 | SAC | DR3 |
|---|---|---|---|---|---|
| Pendulum-Swingup | 1.43H | 1.79H | 1.20H | 1.68H | 1.34H |
| Acrobot-Swingup | 1.46H | 1.83H | 1.20H | 1.71H | 1.36H |
| Hopper-Hop | 1.48H | 1.83H | 1.18H | 1.74H | 1.30H |
| Finger-Spin | 1.48H | 1.81H | 1.25H | 1.85H | 1.29H |
| Average Time | 1.47H | 1.81H | 1.21H | 1.74H | 1.32H |

## I    CLARIFICATION OF $\mathbb{SG}$ DEFINITION

We provide a detailed explanation about $\mathbb{SG}$ in this section. The term $\mathbb{SG}$, standing for 'Stop Gradient', refers to a concept intrinsic to the semi-gradient nature of the Bellman backup, as elaborated in section 9.3 of Sutton & Barto (2018).

To elucidate, we formally define the $\mathbb{SG}$ operator as follows.

**Definition 2** (Stop Gradient). The $\mathbb{SG}$ operator is applied to a differentiable function $f : \mathbb{R}^n \to \mathbb{R}^m$ within the context of gradient-based optimization. When this operator is applied, it generates a new function $\mathbb{SG}(f)$, such that for any input vector $\mathbf{x} \in \mathbb{R}^n$, the output during the forward computation remains the same as that of $f$, i.e., $\mathbb{SG}(f)(\mathbf{x}) = f(\mathbf{x})$. However, during the backward computation phase, where gradients are typically calculated through backpropagation, the gradient of $\mathbb{SG}(f)$ with respect to $\mathbf{x}$ is explicitly set to zero. This is irrespective of the actual gradient of $f$ at $\mathbf{x}$. Formally, this concept is encapsulated by the following equation

$$\forall \mathbf{x} \in \mathbb{R}^n, \quad \mathbb{SG}(f)(\mathbf{x}) = f(\mathbf{x}) \quad \text{and} \quad \nabla_{\mathbf{x}}\mathbb{SG}(f)(\mathbf{x}) = \mathbf{0}. \tag{40}$$

## J    CLARIFICATION ON STATE-ACTION PAIR COVERAGE IN BEER

We discuss the state-action pair coverage of BEER in this section since this affects the regularization of rank.

A critical aspect of our approach is the significance of the reachability of the agent. Specifically, the representation of state-action that are not reachable by any policy is deemed irrelevant. This perspective directly influences our approach to handling state-action pairs in BEER.

- **Sampling Mechanism:** BEER incorporates a uniform sampling strategy from the replay buffer, designed to theoretically encompass the entire spectrum of possible state-action pairs. This approach ensures that our representation accounts for a comprehensive set of these pairs.

- **Exploration Enhancements:** In addition to the uniform sampling, BEER incorporates exploration tricks, such as adding noise to actions, to further ensure extensive coverage of state-action spaces.

**Practical Implementation of Sampling.** In practical terms, our implementation involves extensive sampling iterations, amounting to one million, with each batch comprising 256 samples. This substantial sampling framework is important in enhancing the likelihood of covering all possible state-action pairings, thereby addressing concerns regarding limited coverage. The specifics of this process can be found in Algorithm 1.

## K    CONTRAST WITH PRIOR THEORETICAL WORK

Recent literature discusses how to maximize representation rank in bandit settings and RL settings (Papini et al., 2021a;b; Zhang et al., 2023). We discuss the relationship between our work and the theoretical foundations established in Papini et al. (2021a;b); Zhang et al. (2023). Our approach, while acknowledging the importance of maximizing representation rank, presents a distinct perspective compared to these references.

Our work presents a notable contrast to the contributions of Papini et al. (2021a;b); Zhang et al. (2023). These works focus on selecting (pre-given) representations for decision-making problems while our research centers on learning good representations in DRL. The high representation rank assumption in Papini et al. (2021b;a) cannot inherently bound the cosine similarity derived in our paper. Our approach, grounded in the structure of the Bellman equation and neural networks, uniquely bounds this similarity.

**The diversity assumption in Papini et al. (2021b).** The settings and definitions in Papini et al. (2021b) differ significantly from ours. In Papini et al. (2021b), the definition of representation $\mu(s,a) = <\phi(s,a), \theta>$, where $\mu$ is the true reward vector, $\theta$ is a learnable parameters vector. Thus, $\phi$ is a static representation matrix for computing reward. However, in our settings, $\phi(s,a)$ is a learnable representation for directly computing $Q$ values. Let's first ignore the difference in settings. The answer is No. Their main theory is to achieve a good regret, the optimal representations $\phi^*$ must span the whole representation space $\mathbb{R}^d$, where $d$ is the dimension of representation $\phi$. It's a trivial fact that the rank of $\phi$ cannot additionally bound the mean cosine similarity of its column vectors. For example, consider the matrix

$$A = \begin{pmatrix} 1+x & 1-y \\ 1+y & 1-x \end{pmatrix}.$$

Here, $A$ being full rank ($x^2 - y^2 \neq 0$) does not bound the cosine similarity. The cosine similarity (C) is

$$C = \frac{(1-x)(1+x) + (1-y)(1+y)}{\sqrt{(1-x)^2 + (1-y)^2}\sqrt{(1+x)^2 + (1+y)^2}}, \tag{41}$$

and we know that $\lim_{x \to y} = 1$. In the case of a matrix A with full rank, the cosine similarity can be infinitely close to 1 when $x^2 - y^2 \neq 0$, which demonstrates that the rank cannot inherently bound the cosine similarity.

**The UniSOFT assumption in Papini et al. (2021a).** UniSOFT is a necessary condition for constant regret in any MDP with linear rewards. Specifically, it shows that the span of the representation $\phi_h(s,a)$ over all states and actions that are reachable under any policy $\pi$ is the same as the span of the optimal representation $\phi_h^*(s)$ over all states that have a non-zero probability of being reached under the optimal policy $\pi^*$. This implies that the representation map captures all the necessary information to represent the optimal policy. UniSOFT assumption cannot bound the cosine similarity derived in our paper. Because UniSOFT only considers the span of the representation map. UniSOFT doesn't further consider the interrelationships between column vectors of the representation map, allowing high average cosine similarity (<1) for the column vectors. The UniSOFT condition can be satisfied even with high cosine similarity (can be infinitely close to 1 when $x^2 - y^2 \neq 0$, $A$ is full rank), as it primarily concerns the span of representations.

