# OpenReview forum: "Adaptive Regularization of Representation Rank as an Implicit Constraint of Bellman Equation"
_ICLR.cc/2024/Conference — ICLR 2024 poster_

### Official Review · Reviewer_ef2F · 2023-10-14

**Soundness:** 3 good
**Presentation:** 3 good
**Contribution:** 2 fair
**Rating:** 6
**Confidence:** 2

**Summary:**

This paper studies an important issue of representation rank in deep reinforcement learning (DRL).  To tackle this problem, the authors propose BEER, which leverages the Bellman equation to derive an upper bound on the cosine similarity between consecutive state-action pair representations. This bound is then used as a regularizer to adaptively control the representation rank during training. The empirical studies on DeepMind control tasks validate the high efficiency of the proposed algorithm.

**Strengths:**

The paper is well written and easy to follow. The proposed adaptive regularization is new and effective.

The empirical results clearly validate the high efficiency of BEER, which is impressive. Since I am not an expert in this specific area and haven't followed this line of recent literature, I will refer to others reviewers' opinions on the experiments.

**Weaknesses:**

I have some minor questions:

1. The authors intuitively explained the intrinsic similarity between the cosine similarity and the representation rank with examples. I am curious about if there is some formal statement or proof on this topic, since it serves as a very important property of your paper. There is some parameter in the definition of the representation rank (e.g. $\epsilon$) in Definition 1, but I didn't see it in the following discussion.

2. When the authors transform the constraint in Eqn. (11) to the penalty regularizer with ReLU in Eqn.(12), is there any theoretical support for this transformation? And also is there any theoretical analysis on how to choose the value of $\beta$? I check the ablation study on this hyperparameter on Appendix, but I feel its choice is still quite important but unclear in practice.

For the Eqn. (5), should the $\phi$ be $\Phi$?

**Questions:**

Please refer to the above Weaknesses section.

---

> ### Author Response · Authors · 2023-11-16
> **Response to Reviewer ef2F**
>
> Dear Reviewer ef2F:
>
> We are grateful for your time in reviewing our paper and appreciate your positive remarks about the clarity of our writing and the effectiveness of our adaptive regularization. We address your concerns as follows.
>
> > Q1.1: The authors intuitively explained the intrinsic similarity between the cosine similarity and the representation rank with examples. I am curious about if there is some formal statement or proof on this topic since it serves as a very important property of your paper.
>
> **A1.1**:
> - To the best of our knowledge, there is currently no established formal statement or proof directly correlating cosine similarity with representation rank in the DRL literature. Our work is theoretically motivated.
> - Theoretically, rank cannot strictly bound the cosine similarity without making more further information (e.g. Bellman equation and neural network in our paper).
> - Empirically, researchers [1,2,3] could use cosine similarity to affect the rank. In our work, we explore the influence of cosine similarity on the rank, as detailed in Appendix D.
> - On the one hand, several practices and algorithms in the field [1,2,3] manipulate representation vectors to affect the rank, which is aligned with our approach. For example, [1] maximizes representation rank by aligning learned representation vectors with their initial counterparts.
> - On the other hand, existing theoretical research [4,5,6] about representation rank cannot inherently bound the cosine similarity of the representation vectors.
>
>
> > **Q1.2**: There is some parameter in the definition of the representation rank (e.g. $\epsilon$) in Definition 1, but I didn't see it in the following discussion.
>
> **A1.2**: The parameter $\epsilon$ in Definition 1 aims to filter out small singular values, thereby focusing on vectors that mainly contribute to the rank. We selected $\epsilon=0.01$ for our experiments, as detailed in Tables 2, 3, and 4. This value aligns with the default setting in prior literature (referenced in section C.2 of [1]), providing a standardized basis for comparison and analysis in our work.
>
>
> > Q2.1: When the authors transform the constraint in Eqn. (11) to the penalty regularizer with ReLU in Eqn. (12), is there any theoretical support for this transformation?
>
> **A2.1**:
> - The primal-dual method, typically used for some convex optimization problems, is not feasible here due to the non-convex nature of the original problem in Eqn. (11).  The original problem in Equation 11 involves optimizing $\phi$ and $w$ separately inside. The primal-dual method is still not guaranteed to convergence.
> - Our work is theoretically motivated. Our use of the ReLU function in the regularizer ensures the optimizer does not maximize the representation rank once the upper bound holds. Specifically, when the left-hand side of Equation (11) is less than its right-hand side, the gradients of our regularizer remain consistently zero due to the properties of $\nabla_x \text{ReLU}(x)=0$ for $x<0$.
> - Empirically, according to our analysis in Figure 6 (Appendix E), the $C$ values of BEER empirically ensure the constraint is satisfied. Our method aligns with general practices in DRL literature [1,2,3], where adding a regularizer to the original optimization objective is a common method.
>
>
> > Q2.2: And also is there any theoretical analysis on how to choose the value of $\beta$? I check the ablation study on this hyperparameter on Appendix, but I feel its choice is still quite important but unclear in practice.
>
> **A2.2**:
> - Theoretical insights: The choice of $\beta$ determines the relative impact of the two losses (primary optimization and regularization) on parameter gradients. To avoid undermining the primary optimization objective, the regularizer's loss should be considerably smaller than the primary loss.
> - Empirically, our experiments in Appendix F indicate that a relatively small $\beta$ works well. $\beta$ varying over a small domain of values ([0.0005,0.001] in our experiments) results in equally good performance.
> - P.S. One of the reasons why we selected a consistent, fixed $\beta$ across all the experiments is to fairly evaluate our algorithm considering the recent crisis in the model evaluation [7].
>
>
> > Q3: For the Eqn. (5), should the $\phi$ be $\Phi$?
>
> **A3.** Thank you for highlighting this typographical error. We have modified it in the revised manuscript, where $\phi$ is now correctly denoted as $\Phi$.
>
> We hope our responses provide clarity on the issues raised. We appreciate your feedback and believe that it will greatly enhance the quality and impact of our paper. Thanks for considering our response.
>
> Sincerely,
> Authors of submission 1504.

---

> > ### Author Response · Authors · 2023-11-16
> > **References**
> >
> > References
> >
> > [1] Lyle C, Rowland M, Dabney W. Understanding and Preventing Capacity Loss in Reinforcement Learning[C]//International Conference on Learning Representations. 2021.
> >
> > [2] Kumar A, Agarwal R, Ma T, et al. DR3: Value-Based Deep Reinforcement Learning Requires Explicit Regularization[C]//International Conference on Learning Representations. 2021.
> >
> > [3] He Q, Su H, Zhang J, et al. Frustratingly Easy Regularization on Representation Can Boost Deep Reinforcement Learning[C]//Proceedings of the IEEE/CVF Conference on Computer Vision and Pattern Recognition. 2023: 20215-20225.
> >
> > [4] Papini, Matteo, et al. "Leveraging good representations in linear contextual bandits." International Conference on Machine Learning. PMLR, 2021.
> >
> > [5] Papini, Matteo, et al. "Reinforcement learning in linear mdps: Constant regret and representation selection." Advances in Neural Information Processing Systems 34 (2021): 16371-16383.
> >
> > [6] Zhang, Weitong, et al. "Provably efficient representation selection in low-rank Markov decision processes: from online to offline RL." Uncertainty in Artificial Intelligence. PMLR, 2023.
> >
> > [7] Henderson P, Islam R, Bachman P, et al. Deep reinforcement learning that matters[C]//Proceedings of the AAAI conference on artificial intelligence. 2018, 32(1).

---

> > > ### Author Response · Authors · 2023-11-19
> > > **Looking forward to further discussion**
> > >
> > > Dear Reviewer ef2F,
> > >
> > > Thank you for your valuable feedback. In our response, we provided clarifications on the intrinsic relationship between cosine similarity and representation rank, the choice of parameter $\epsilon$ in our definition, the theoretical insight for our regularization transformation, and the insight of the selection of $\beta$.
> > >
> > > Your comments have significantly improved the clarity of our manuscript. We are looking forward to addressing any additional concerns you may have.
> > >
> > > Sincerely,
> > >
> > > Authors of Submission 1504

---

> > > > ### Comment · Reviewer_ef2F · 2023-11-22
> > > > **Thank you for your response**
> > > >
> > > > Thank you for your detailed response to all my questions. After reading the rebuttal and feedbacks from other reviewers, I feel the paper is on the borderline, and hence I'd like to keep my score.

---

> ### Author Response · Authors · 2023-11-22
> **Thanks for your reply!**
>
> Dear Reviewer ef2F,
>
> Thank you for acknowledging our detailed responses to your insightful comments.  We would respectfully appreciate it if you could reevaluate the confidence score, considering the points highlighted below:
>
> - Addressing your concerns. We provide comprehensive responses to address the concerns raised in your review. Our revised manuscript and responses reflect a thorough consideration of your insightful feedback. Your insightful comments have significantly strengthened our work.
>
> - Contribution to the community. Our work introduces a novel adaptive method to control the representation rank in deep reinforcement learning, potentially offering new directions for future research. We believe this aspect enhances the paper's value to the community.
>
> We hope these points might offer a new perspective on the potential impact of our paper. Thanks for your reply again.
>
> Sincerely,
>
> Authors of Submission 1504

---

### Official Review · Reviewer_dvZZ · 2023-10-29

**Soundness:** 3 good
**Presentation:** 3 good
**Contribution:** 2 fair
**Rating:** 6
**Confidence:** 4

**Summary:**

This paper studies an implicit regularization of representation rank in deep RL. The authors draw the intuition between the representation rank and the cosine similarity between adjustment state-action pairs.

**Strengths:**

- This paper is well written and easy to follow
- The experiment shows a good performance and backup the theoretical insights

**Weaknesses:**

- The argument that the cosine similarity of the ** adjustment** state-action pairs $(s, a)$ and $(s', a')$ does not rigorously lead to the rank. This is because one can only control the similarity between limited state-action pairs, instead of all possible state-action pairs. Think of an extreme case that the representation in adjustment state-action pairs is iterating between $(0, 1, 0, \cdots, 0)$ and $(1, 0, 0, \cdots, 0)$, the implicit regularization will fail.
- It seems that in the experiment (Figure 1), the representation rank of the proposed method is dropping. It would be beneficial if the authors can provide some explanation on this

**Questions:**

- Maximizing the representation rank has always been an important part of the literature. Several theoretical works [1, 2, 3] show that as long as all the representations are in the span of the covariance matrix (i.e. $x \in E[xx^\top], the diversity assumption in [1] or the UniSOFT assumption in [2]), the performance can be improved. It would be beneficial for authors to comment on the relationship between this paper and these theoretical results, e.g. will the cosine similarity be well-bounded under these assumptions?

[1] Papini, Matteo, et al. "Leveraging good representations in linear contextual bandits." International Conference on Machine Learning. PMLR, 2021.
[2] Papini, Matteo, et al. "Reinforcement learning in linear mdps: Constant regret and representation selection." Advances in Neural Information Processing Systems 34 (2021): 16371-16383.
[3] Zhang, Weitong, et al. "Provably efficient representation selection in low-rank Markov decision processes: from online to offline RL." Uncertainty in Artificial Intelligence. PMLR, 2023.

---

> ### Author Response · Authors · 2023-11-16
> **Response to Reviewer dvZZ (1/2)**
>
> Dear Reviewer dvZZ:
>
> We are grateful for your comprehensive review and constructive feedback on our submission. Your recognition of the clarity of our writing and the effectiveness of our experimental results is highly appreciated. We address your concerns as follows.
>
> > Q1: The argument that the cosine similarity of the ** adjustment** state-action pairs $(s,a)$ and $(s',a')$ does not rigorously lead to the rank. This is because one can only control the similarity between limited state-action pairs, instead of all possible state-action pairs. Think of an extreme case that the representation in adjustment state-action pairs is iterating between $(0,1,0,\cdots,0)$ and $(1,0,0,\cdots,0)$, the implicit regularization will fail.
>
> **A1.** We thank you for pointing out this critical aspect! Our design indeed aims to access the entire spectrum of possible state-action pairs. As highlighted in [2], "In RL, the reachability of a state plays a fundamental role. For example, representation of states that are not reachable by any policy are irrelevant". The unreachable state-action pairs are irrelvant for the numerical representation rank (in Definition 1) because they cannot be sampled. We humbly hope to draw your attention to our sampling process as detailed in Algorithm 1, lines 7 to 10.
>
> - **Sampling mechanism allows access to all possible (s,a) pairs.**: BEER employs a uniform sampling strategy from the replay buffer, which theoretically allows access to the entire spectrum of state-action pairs. This strategy is a fundamental aspect of our methodology, ensuring a comprehensive representation of the state-action pairs. Additionally, the exploration tricks in BEER, which introduce noise to actions, further ensure the agent covers as many state-action pairs as possible.
>
>
> - **Practical Implementation**: We implement this with substantial sampling iterations – one million – with each batch comprising 256 samples. This extensive sampling framework significantly enhances the likelihood of covering all possible state-action pairings, thereby mitigating concerns about limited coverage.
>
> We have added more details in our revised manuscript (Appendix J) to clarify these points and highlight the robustness of our sampling process.
>
>
>
>
> >Q2: It seems that in the experiment (Figure 1), the representation rank of the proposed method is dropping. It would be beneficial if the authors can provide some explanation on this.
>
>
> **A2.**  The representation rank in Figure 1 is indicative of BEER's adaptive regularization capability. A constant high or low rank is impractical for different tasks. Our method adaptively adjusts the rank to balance the model, preventing overfitting and underfitting. The criterion for evaluating rank is the approximation error – the absolute difference between estimated and true values. BEER's lower approximation error compared to other algorithms in Figure 1 demonstrates its effectiveness in achieving a preferable rank. This adaptive regularization is important to BEER's performance improvement.

---

> > ### Author Response · Authors · 2023-11-16
> > **Response to Reviewer dvZZ (2/2)**
> >
> > > **Q3**: Maximizing the representation rank has always been an important part of the literature. Several theoretical works [1, 2, 3] show that as long as all the representations are in the span of the covariance matrix (i.e. $x \in E[xx^\top], the diversity assumption in [1] or the UniSOFT assumption in [2]), the performance can be improved. It would be beneficial for authors to comment on the relationship between this paper and these theoretical results, e.g. will the cosine similarity be well-bounded under these assumptions?
> >
> > A3. Difference.** Our work presents a notable contrast to the contributions of [1, 2, 3]. These works focus on selecting (pre-given) representations for decision-making problems while our research focuses on learning good representations in DRL. The high representation rank assumption in [1,2] **cannot** inherently bound the cosine similarity derived in our paper. Our approach, grounded in the structure of the Bellman equation and neural networks, uniquely bounds this similarity. We discuss more details as follows.
> >
> >
> > - **The diversity assumption in [1].** The settings and definitions in [1] differ significantly from ours. In [1], the definition of representation $\mu(s,a)=<\phi(s,a), \theta>$, where $\mu$ is the true reward vector, $\theta$ is a learnable parameters vector. Thus, $\phi$ is a **static** representation matrix for computing the reward. However, in our settings, $\phi(s,a)$ is a learnable representation for directly computing $Q$ values.
> >
> >     - Should the diversity assumption in [1] bound the similarity in our paper? Let's first ignore the difference in settings. The answer is No. Their main theory is to achieve a good regret, the optimal representations $\phi^*$ must span the whole representation space $\mathbb{R}^d$, where $d$ is the dimension of representation $\phi$. It's a trivial fact that the rank of $\phi$ cannot additionally bound the mean cosine similarity of its column vectors. For example, consider the matrix
> > $$
> > A = \begin{pmatrix}
> > 1+x & 1-y \\\\
> > 1+y & 1-x
> > \end{pmatrix}.
> > $$
> >
> > Here, $A$ being full rank ($x^2-y^2 \neq 0$) does not bound the cosine similarity.
> > The cosine similarity (C) is
> > $$
> > C = \frac{(1 - x)(1 + x) + (1 - y)(1 + y)}{\sqrt{(1 - x)^2 + (1 - y)^2} \sqrt{(1 + x)^2 + (1 + y)^2}},
> > $$
> > and we know that $\lim_{x\to y} = 1$. In the case of a matrix A with full rank, the cosine similarity can be infinitely close to 1 when $x^2-y^2 \neq 0$, which demonstrates that the rank does not inherently bound the cosine similarity.
> >
> > - **The UniSOFT assumption in [2].** Should the UniSOFT assumption in [2] bound the similarity in our paper? No.
> >
> >     - UniSOFT is a necessary condition for constant regret in any MDP with linear rewards. Specifically, it shows that the span of the representation $\phi_h(s, a)$ over all states and actions that are reachable under any policy $\pi$ is the same as the span of the optimal representation $\phi^*_h(s)$ over all states that have a non-zero probability of being reached under the optimal policy $\pi^*$. This implies that the representation map captures all the necessary information to represent the optimal policy.
> >     - UniSOFT assumption cannot bound the cosine similarity derived in our paper. This is because UniSOFT only considers the span of the representation map. UniSOFT doesn't further consider the interrelationships between column vectors of the representation map, allowing high average cosine similarity (<1) for the column vectors. The UniSOFT condition can be satisfied even with high cosine similarity (e.g. can be infinitely close to 1 when $x^2-y^2 \neq 0$, $A$ is full rank), as it primarily concerns the span of representations.
> >
> > In our revised submission, we have added a new Appendix K to discuss these theoretical perspectives and their relation to our study, and we have also cited these works in the 'Related Work' section to provide a more comprehensive literature review.
> >
> >
> > We hope our responses clarify the claims in your concerns and demonstrate the rigor of our approach. We are committed to enhancing the clarity and depth of our paper. Thank you for considering our response.
> >
> > Thank you again for your valuable feedback, which we believe significantly strengthens our paper.
> > Best regards,
> > Authors of submission 1504.
> >
> > **References**
> >
> > [1] Papini, Matteo, et al. "Leveraging good representations in linear contextual bandits." International Conference on Machine Learning. PMLR, 2021.
> >
> > [2] Papini, Matteo, et al. "Reinforcement learning in linear mdps: Constant regret and representation selection." Advances in Neural Information Processing Systems 34 (2021): 16371-16383.
> >
> > [3] Zhang, Weitong, et al. "Provably efficient representation selection in low-rank Markov decision processes: from online to offline RL." Uncertainty in Artificial Intelligence. PMLR, 2023.

---

> > > ### Author Response · Authors · 2023-11-19
> > > **Looking forward to further discussion**
> > >
> > > Dear Reviewer dvZZ,
> > >
> > > We appreciate your thorough review and insightful comments. In our response, we addressed your concerns about the control of state-action pair similarities and the dropping representation rank observed in Figure 1. We also provided a detailed discussion on the relationship of our work with existing theoretical frameworks, emphasizing the unique aspects of our approach.
> > >
> > > We understand the importance of a rigorous theoretical foundation and empirical justification in DRL research. If you have any further questions or require to discuss the theoretical/empirical aspects in more detail, we would be happy to participate and refine our work accordingly.
> > >
> > > Best regards,
> > >
> > > Authors of Submission 1504

---

### Official Review · Reviewer_Joqj · 2023-10-30

**Soundness:** 3 good
**Presentation:** 3 good
**Contribution:** 3 good
**Rating:** 6
**Confidence:** 3

**Summary:**

This work studies how to learn a low-rank representation in reinforcement learning (RL). Following several previous works, the authors claimed that RL favors a representation with moderate rank, and they proposed a new regularization method based derived from the Bellman equation. In detail, such a regularization method controls the complexity of the learned representation, and it encourages the rank of the representation to be small empirically. Combined with the regularization method, several baseline methods such as DQN and DDPG outperform existing baselines.

**Strengths:**

1. The presentation of this paper is clear.
2. The experiment setup is described very clear.
3. The literature review is complete.

**Weaknesses:**

There are several things that can improve the paper. For instance,
1. On page 4, the definition of $\bar{\phi}(s',a')$ seems not clear since $(s',a')$ serves as both the input of $\bar{\phi}$ and the random variable which is going to be integrated.
2. On page 5, what is the exact definition of $\mathbb{SG}$?

I do not quite get why the authors need to design the regularization term as in (12). It seems more natural to me to set the regularization term as in (19), which avoids a calculation of the gradient of an inverse term $1/|\phi(s,a)|$ which might hurt the stability of the optimization process.

The logic behind the superior performance of BEER is not clear to me. The authors tried to claim that 'representation rank affects the model performance (approximation error), while BEER explicitly controls the rank, thus BEER outperforms other baseline methods'. However, according to figure 2, InFeR has a similar representation rank as BEER, while it performs worse than BEER. Is there any explanation why such a phenomenon happens? Meanwhile, in simpler tasks (Grid World), the rank of BEER is higher than other baselines, while in complex tasks (Lunan radar), the rank of BEER is lower than other baselines. Due to the inconsistency, it is doubtful whether the performance gain is due to the better control of representation rank. If so, it indeed deserves more explanation.

**Questions:**

See Weaknesses.

---

> ### Author Response · Authors · 2023-11-16
> **Response to Reviewer Joqj (1/2)**
>
> Dear Reviewer Joqj,
>
> Thank you for your insightful feedback and the time you have invested in reviewing our paper. Your comments significantly improve the quality of our paper. We appreciate your acknowledgment of the clear presentation, detailed experimental setup, and comprehensive literature review in our work. We address your concerns as follows.
>
>
> > **Q1**: On page 4, the definition of $\bar{\phi}(s',a')$ seems not clear since $(s',a')$ serves as both the input of $\bar{\phi}$ and the random variable which is going to be integrated.
>
> **A1**. Thank you for pointing out this! We have revised the notation from $\bar{\phi}(s',a')$ to $\overline{\phi(s',a')}$, where $\overline{\phi(s',a')} = \mathbb{E}_{s',a'} \phi(s',a')$. This modification does not affect the theoretical correctness and rigor of the original text.
>
>
> > **Q2**: On page 5, what is the exact definition of $\mathbb{SG}$?
>
>
> **A2**. The term $\mathbb{SG}$, denoting 'Stop Gradient', is a concept from the semi-gradient nature of the Bellman backup, as discussed in Section 9.3 of [1].
>
> To provide a more precise understanding, $\mathbb{SG}$ can be defined as follows:
>
> $\textbf{Definition 2: Stop Gradient.}$ The $\mathbb{SG}$ operator is applied to a differentiable function $f: \mathbb{R}^n \rightarrow \mathbb{R}^m$ within the context of gradient-based optimization. When this operator is applied, it generates a new function $\mathbb{SG}(f)$, such that for any input vector $\mathbf{x} \in \mathbb{R}^n$, the output during the forward computation remains the same as that of $f$, i.e., $\mathbb{SG}(f)(\mathbf{x}) = f(\mathbf{x})$. However, during the backward computation phase, where gradients are typically calculated through backpropagation, the gradient of $\mathbb{SG}(f)$ with respect to $\mathbf{x}$ is explicitly set to zero. This is irrespective of the actual gradient of $f$ at $\mathbf{x}$. Formally, this concept is encapsulated by the following equation
>
> $$
> \forall \mathbf{x} \in \mathbb{R}^n, \quad \mathbb{SG}(f)(\mathbf{x}) = f(\mathbf{x}) \quad \text{and} \quad \nabla_{\mathbf{x}} \mathbb{SG}(f)(\mathbf{x}) = \mathbf{0}.
> $$
>
> We have included this analysis in Appendix I of our revised manuscript.
>
>
> > **Q3**: I do not quite get why the authors need to design the regularization term as in (12). It seems more natural to me to set the regularization term as in (19), which avoids a calculation of the gradient of an inverse term $1/\phi(s,a)$ which might hurt the stability of the optimization process.
>
>
> **A3**. We apologize for the unintentional missing of a $\mathbb{SG}$ symbol in Eq 12. The corrected equation should read as follows:
>
> $$
> \mathcal{R}(\theta) = \text{ReLU}\Big( \cos(\phi(s,a),\mathbb{SG}\overline{\phi(s',a')})  -  \mathbb{SG}\big((\| \phi(s,a) \|^2 + \gamma^2\| \overline{\phi}(s',a') \|^2 - \frac{\|r\|^2}{\|w\|^2} )\frac{1}{2\gamma \| \phi(s,a) \| \| \overline{\phi(s',a')} \| } \big)\Big),
> $$
>
> where the $\mathbb{SG}$ operation is applied to the derived upper bound. Note that our experimental results are based on this corrected equation (demonstrated by the submitted code) rather than the original Eq. (12). As a result of the $\mathbb{SG}$ operation, the gradient of $1/\phi(s,a)$ is not computed, thereby preserving the stability of the optimization process.

---

> > ### Author Response · Authors · 2023-11-16
> > **Response to Reviewer Joqj (2/2)**
> >
> > > **Q4.1.** The logic behind the superior performance of BEER is not clear to me. The authors tried to claim that 'representation rank affects the model performance (approximation error), while BEER explicitly controls the rank, thus BEER outperforms other baseline methods'. **Q4.2.** However, according to figure 2, InFeR has a similar representation rank as BEER, while it performs worse than BEER. Is there any explanation why such a phenomenon happens? Meanwhile, in simpler tasks (Grid World), the rank of BEER is higher than other baselines, while in complex tasks (Lunan radar), the rank of BEER is lower than other baselines. Due to the inconsistency, it is doubtful whether the performance gain is due to the better control of representation rank. If so, it indeed deserves more explanation.
> >
> >
> > **A4.1 (Logic).**  BEER's effectiveness lies in its adaptive regulation of representation rank. Different tasks require different representation ranks. Overly high ranks may lead to overfitting, while too low ranks can cause underfitting. BEER's principle is based on the Bellman equation and adaptively balances this rank to optimize the performance.
> >
> >
> > **A4.2 (Experiments).** We highlight the fact that different tasks demand different representation ranks. Consistently high or consistently low representation rank is **not optimal** for all the tasks.
> >
> >
> >
> > Determining whether a rank is good or bad is challenging. An effective guideline is that a rank is preferred if it reduces the approximation error. We thus utilize approximation error as our primary criterion for rank and model evaluation. This metric directly and precisely assesses the quality of ranks and models.
> >
> > Our experiments show that BEER consistently achieves lower approximation errors, leading to superior performance. BEER's representation rank, as depicted in Figures 1(b) and 2(b), demonstrates its efficacy in maintaining ranks adaptive to different tasks, which is critical to its performance.
> >
> >
> > Regarding Figure 2, the ranks of BEER and InFeR are statistically different, as indicated by their non-overlapping confidence intervals.
> >
> > We hope these clarifications address your concerns. We remain open to any additional queries and are committed to continually improving our work. Thank you for considering our response.
> >
> >
> > Sincerely,
> > Authors of Submission 1504
> >
> >
> >
> > **References**
> >
> > [1] Sutton R S, Barto A G. Reinforcement learning: An introduction[M]. MIT press, 2018. Section 9.3.

---

> > > ### Author Response · Authors · 2023-11-19
> > > **Looking forward to further discussion**
> > >
> > > Dear Reviewer Joqj,
> > >
> > > Thank you for your insightful feedback. In our response, we revised the notation for clarity, defined the 'Stop Gradient' term, and corrected Equation 12 to address your concerns. We also provided a detailed explanation of BEER's effectiveness and its adaptive regulation of representation rank, emphasizing its task-specific performance benefits.
> > >
> > > We value your insightful comments on our method and are open to further discussions to improve our manuscript. If you have any additional concerns or require more detailed explanations, please feel free to ask, and we will be more than happy to provide further responses.
> > >
> > > Best regards,
> > >
> > > Authors of Submission 1504

---

### Official Review · Reviewer_ykJX · 2023-11-02

**Soundness:** 3 good
**Presentation:** 3 good
**Contribution:** 3 good
**Rating:** 6
**Confidence:** 2

**Summary:**

The paper proposes a novel method to control the representation rank of neural networks in deep reinforcement learning (DRL), which measures the expressive capacity of value networks. They argue that existing methods either ignore or unboundedly maximize the representation rank, which can lead to overfitting or underfitting problems.

**Strengths:**

The proposed method is well-founded. The authors establish an upper bound on the cosine similarity between the representations of consecutive state-action pairs, using the Bellman equation1. They demonstrate that this bound indirectly restricts the rank of the representation and offers a criterion for adaptive control.

**Weaknesses:**

It is recommended to thoroughly investigate the computational overhead of the proposed method.

**Questions:**

See Weaknesses.

---

> ### Author Response · Authors · 2023-11-16
> **Response to Reviewer ykJX**
>
> Dear Reviewer ykJX,
>
> Thank you very much for your detailed review and insightful comments on our manuscript. Your feedback has been instrumental in enhancing the quality of our work.  We appreciate your recognition of the foundational aspects of our method and its capability in controlling the representation rank in DRL.
>
> In response to your concerns about the computational overhead of BEER, we would like to offer the following clarification.
>
> > **Q1.** It is recommended to thoroughly investigate the computational overhead of the proposed method.
>
>
> **A1.** BEER introduces an additional loss term but maintains the original network architecture. This significantly mitigates the increase in computational burden. To provide a more detailed evaluation, we have conducted two key analyses:
>
>
> 1. **The number of learnable parameters**. BEER operates as a regularizer without introducing any new learnable parameters to the base algorithm. This contrasts with methods like InFeR, which incorporates multiple additional heads to the value networks and significantly increases the number of learnable parameters.
>
> 2. **Runtime analysis.** BEER’s runtime is competitive with other algorithms and, in some cases, even more efficient. We have carefully compared the runtime of BEER with other algorithms focusing on representation rank, including SAC, using the same standard evaluation protocol. The experiments were executed on a GPU server equipped with an Intel(R) Xeon(R) Gold 6240 CPU @ 2.60GHz and NVIDIA GeForce RTX 2080 Ti GPUs. Each task was run sequentially to ensure a fair comparison. The results are as follows (H=hours).
>
>
>
> Table 1. Runtime comparison of BEER and other algorithms. The runtime of BEER is competitive with other algorithms.
>
> | Env                | BEER  | INFER | TD3   | SAC   | DR3   |
> |--------------------|-------|-------|-------|-------|-------|
> | pendulum-swingup   | 1.43H | 1.79H | 1.20H | 1.68H | 1.34H |
> | acrobot-swingup    | 1.46H | 1.83H | 1.20H | 1.71H | 1.36H |
> | hopper-hop         | 1.48H | 1.83H | 1.18H | 1.74H | 1.30H |
> | finger-spin        | 1.48H | 1.81H | 1.25H | 1.85H | 1.29H |
> | **Average Time**   | 1.47H | 1.81H | 1.21H | 1.74H | 1.32H |
>
>
> These results show that BEER's runtime is competitive with other algorithms.
>
> We have included this analysis in Appendix H of our revised manuscript for a comprehensive understanding.
>
> We hope this response addresses your concerns and further supports the feasibility of our approach. We remain open to any additional queries and are committed to continually improving our work. Thank you for considering our response.
>
>
> Sincerely,
> Authors of Submission 1504.
>
> References
>
> [1] Lyle C, Rowland M, Dabney W. Understanding and Preventing Capacity Loss in Reinforcement Learning[C]//International Conference on Learning Representations. 2021.

---

> > ### Author Response · Authors · 2023-11-19
> > **Looking forward to further discussion**
> >
> > Dear Reviewer ykJX,
> >
> > We sincerely thank you for your constructive feedback. In our response, we addressed your concern about the computational overhead of BEER by comparing its runtime and number of learnable parameters with other algorithms. As highlighted in our revised Appendix H, these analyses demonstrate BEER's efficiency.
> >
> > We appreciate your emphasis on this aspect. If you have further insights or require further clarification, we are ready to discuss and incorporate your valuable suggestions.
> >
> >
> > Best regards,
> >
> > Authors of submission 1504

---

### Author Response · Authors · 2023-11-16
**General Response**

Dear Reviewers and AC,


Thank you each for your work, detailed reviews, and the positive feedback on our manuscript. We are grateful for your appreciation of the clarity of our writing, the robustness of our methodology, and the depth of our experimental analysis.


We have carefully considered and individually addressed each of your concerns in our revised submission. Your insights have been invaluable in enhancing the overall quality of our work.


Sincerely,
Authors of Submission 1504.

---

### Author Response · Authors · 2023-11-22
**A gentle reminder for the close of the author-reviewer discussion & summary of the rebuttal**

Dear Reviewers and AC,

As the author-reviewer discussion period is closing at the end of Wednesday, Nov 22nd (AOE), we would like to call for any further discussion or comments on our submission.


In our rebuttal, we addressed the following raised concerns/misunderstandings. Corresponding changes are marked in red in the revised submission.

- Computational Efficiency (@Reviewer ykJX): We clarified the computational aspects of our BEER algorithm and provided comparative runtime analyses.
- Clarifications on Definitions and Equations (@R Joqj): We revised our manuscript for better clarity, particularly concerning definitions and equations.
- Performance and Representation Rank (@R Joqj, dvZZ): We offered insights into BEER's adaptive regulation of representation rank, crucial for diverse tasks.
- Theoretical and Empirical Justifications (@R dvZZ): We provided theoretical justification and empirical explanation for the correlation between representation rank and performance.
- Distinct Theoretical Contributions (@R dvZZ): We highlighted how our work diverges from the existing theoretical literature, emphasizing our novel approach.
- Parameter Choices and Theoretical Insight (@R ef2F): We explained the logic behind our parameter choices and the transformations.

 We believe that these clarifications and additional details strengthen our paper and address the reviewers' concerns.

We understand the constraints of time and workload that reviewers and AC face, and we appreciate the effort already put into evaluating our work. If there are any additional insights, questions, or clarifications on our responses/submission that you would like to discuss with us, we would be very grateful to hear them. Your feedback is invaluable for the improvement of our research.


Best regards,

Authors of submission 1504.

---

### Meta-Review · Area_Chair_JDUJ · 2023-12-16

**Metareview:**

This paper addresses the challenge of optimizing representation rank in Deep Reinforcement Learning (DRL). Introducing the Bellman equation-based automatic rank Regularizer (BEER), the approach adapts representation rank using a theoretical foundation derived from the Bellman equation. Validation through experiments showcases BEER's effectiveness in improving DRL agent performance, outperforming baselines across complex continuous control tasks.

The paper was borderline but the results are significant. The authors also seem to have addressed the reviewers' concerns adequately in the rebuttal, so we accept the submission. Please incorporate all the comments and clarifications in the final paper.

**Justification For Why Not Higher Score:**

Borderline

**Justification For Why Not Lower Score:**

Significant evaluation

---

### Decision · Program_Chairs · 2024-01-16

Accept (poster)